# Do different prompting methods yield a common task representation in language models?

**Guy Davidson[1], Todd M. Gureckis[2], Brenden M. Lake[3], Adina Williams[1]**
[1]FAIR at Meta, [2]New York University, [3]Princeton University
`guy.davidson@nyu.edu`
`todd.gureckis@nyu.edu, brenden@princeton.edu`
`adinawilliams@meta.com`

## Abstract

Demonstrations and instructions are two primary approaches for prompting language models to perform in-context learning (ICL) tasks. Do identical tasks elicited in different ways result in similar representations of the task? An improved understanding of task representation mechanisms would offer interpretability insights and may aid in steering models. We study this through *function vectors* (FVs), recently proposed as a mechanism to extract few-shot ICL task representations. We generalize FVs to alternative task presentations, focusing on short textual instruction prompts, and successfully extract instruction function vectors that promote zero-shot task accuracy. We find evidence that demonstration- and instruction-based function vectors leverage different model components, and offer several controls to dissociate their contributions to task performance. Our results suggest that different task prompting forms do not induce a common task representation through FVs but elicit different, partly overlapping mechanisms. Our findings offer principled support to the practice of combining instructions and task demonstrations, imply challenges in universally monitoring task inference across presentation forms, and encourage further examinations of LLM task inference mechanisms.

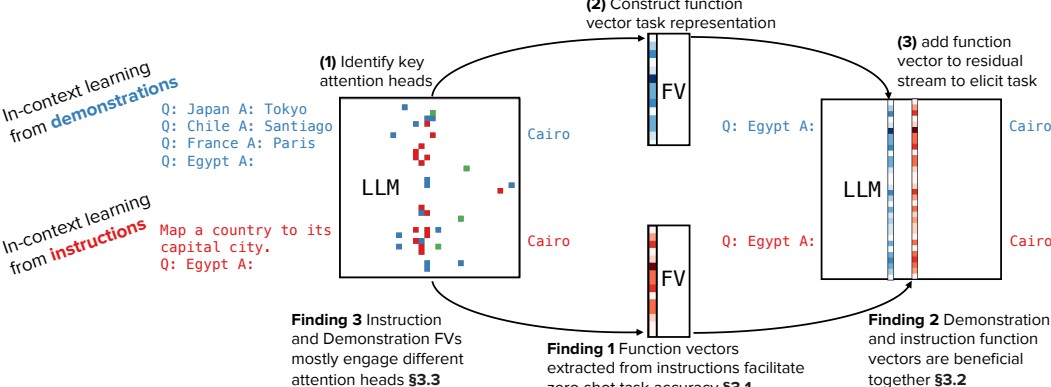

Figure 1: **Language model task representations depend on the form of task presentation**. We compare in-context learning task representations formed from demonstrations with those formed from instructions using function vectors (FVs). The process of extracting FVs is shown in Steps (1)-(3). We highlight several findings: We successfully extract FVs from instructions (§3.1); Instruction FVs offer complementary benefits when applied with demonstration FVs (§3.2); Different prompting methods yield distinct task representations (§3.3; highlighted squares on the left LLM are Llama-3.1-8B-Instruct attention heads: those identified by demonstrations only, those identified by instructions only, and shared ones; columns are layers, rows are head indices, see Figure 2D).

39th Conference on Neural Information Processing Systems (NeurIPS 2025).

# 1 Introduction

If you prompt a large language model (LLM) with in-context examples "Q: Japan A: Tokyo Q: Chile A: Santiago Q: France A: Paris Q: Egypt A:" or with instructions "Map a country to its capital city: Q: Egypt A:", you expect to get the same answer (Figure 1). These two prompts share only the final query in common, but imply the same underlying task for the network to perform. This paper explores how these two prompting methods result in similar or different representations of the specified task in the language model. We consider these representations *task representations*, as they reflect information in the network that induces the correct mapping for a given task, rather than the answer itself.

We focus on two prevalent approaches to specifying tasks to LLMs: demonstrations and instructions. The ability of LLMs to perform in-context learning (ICL) from demonstrations has been of considerable interest since GPT-3 (Brown et al., 2020; Lampinen et al., 2024). With appropriate fine-tuning, language models can also follow textual instructions, facilitating a far broader range of use cases (Chung et al., 2022). We follow a considerable literature studying the mechanisms that govern ICL (e.g., Olsson et al., 2022; Chen et al., 2024; Akyürek et al., 2024) and those that promote instruction-following (Stolfo et al., 2024; Wu et al., 2024). Prior work studied either ability by itself; here, we examine to what extent the two share representations and mechanisms.

We approach this question by leveraging a new interpretability method known as *function vectors* (FVs, Todd et al., 2024). FVs are *causal* patterns identified from the intermediate layers of a language model that mediate its ability to perform a task. Todd et al. demonstrate that FVs, as a single additive intervention to a model's latent activity, successfully cause the execution of a task in a different (or empty) context. We extend Todd et al.'s FV extraction method from using specifically in-context demonstrations to any form of task presentation, including instruction prompts. We then compare the representations elicited by demonstrations and textual instructions, examining their effectiveness at inducing task-following behavior, latent activity similarity, and elicited internal mechanisms.

Table 1 summarizes our key findings. Our extension of the function vector identification procedure successfully extracts instruction function vectors, and these promote zero-shot task accuracy (§3.1). We validate that demonstration and instruction FVs contain complementary information by intervening with both simultaneously, and find it conveys task performance benefits beyond using either FV alone (§3.2). Next, we examine the attention heads identified in demonstration and instruction FVs, and find that most are identified by one type of FV, with only a few shared by both (§3.3). We evaluate the functional implication of the different attention heads and find an asymmetry between the relevance of demonstration-identified attention heads to instructions and instruction-identified heads to demonstrations (§3.4). Finally, noting that instruction FVs function better in post-trained models, we find that we can steer base models with post-trained model instruction FVs, and we identify the relevant post-training stages for instruction FV extraction (§3.5).

Overall, our results suggest that **different task presentations do not induce a common task representation through function vectors, but activate partly overlapping mechanisms and induce jointly beneficial representations.** Our findings provide new insights into how LLMs represent tasks and offer an explanation for why combining instructions and in-context examples often improves model performance.

Table 1: Guiding research question and summary of findings

|  | Claim | Section |
|---|---|---|
| **Question** | Do different ways of presenting the same task elicit a common task representation? | |
| **Method** | Extend Function Vectors (FVs) from in-context demonstrations to instructions (and other) task presentations, and analyze their properties. | |
| **Finding 1** | Function vectors extracted from instructions facilitate zero-shot task accuracy. | 3.1 |
| **Finding 2** | Demonstration and instruction FVs are beneficial together. | 3.2 |
| **Finding 3** | Instruction and demonstration FVs mostly engage different attention heads. | 3.3 |
| **Finding 4** | Instruction-identified attention heads are more useful for building demonstration FVs than vice versa. | 3.4 |
| **Finding 5** | Instruction FVs from post-trained models can steer base models and arise from supervised fine-tuning and preference optimization. | 3.5 |

## 2 Methods

We describe the function vector extraction procedure outlined by Todd et al. (2024) and our extension from in-context demonstrations to arbitrary task prompts (see Todd et al., 2024, for additional details). The procedure identifies a small set of causally relevant heads in a model $f$ for performing a given task $t$, and uses them to compute the function vector $v_t \in \mathbb{R}^{d_{model}}$, where $d_{model}$ is the model's latent dimension. Each task $t \in \mathcal{T}$ (the set of tasks considered) consists of a supervised dataset $\mathcal{D}_t = \{(x_1, y_1), (x_2, y_2), \cdots, (x_{N_t}, y_{N_t})\}$, where $x_i$ and $y_i$ are (tokenized) string inputs and outputs respectively. We construct $K$-shot in-context demonstration prompts $p_i^t$ for a query example $(x_{iq}, y_{iq})$ as $p_i^t = [(x_{i1}, y_{i1}), (x_{i2}, y_{i2}), \cdots, (x_{iK}, y_{iK}), x_{iq}]$. To identify causally relevant attention heads, consider a set of prompts $P_t$ for a given task $t$ on which a model $f$ succeeds in predicting $y_{iq}$ from $p_i^t$, $P_t = \{p_1^t, p_2^t, \cdots, p_N^t\}$. Denote the output of attention head $a_{lj}$ (head $j$ in layer $l$) in processing the final token of prompt $p_i^t$ as $a_{lj}(p_i^t) \in \mathbb{R}^{d_{model}}$ (projected to the model's latent dimension). We compute the mean task-conditioned activation of each attention head, $\bar{a}_{lj}^t = \frac{1}{|P_t|} \sum_{p_i^t \in P_t} a_{lj}(p_i^t)$. Next, we construct uninformative baseline in-context demonstration prompts $\tilde{p}_i^t$ by shuffling the labels $\tilde{y}_{ik}$ assigned to each $\tilde{x}_{ik}$: $\tilde{p}_i^t = [(x_{i1}, \tilde{y}_{i1}), (x_{i2}, \tilde{y}_{i2}), \cdots, (x_{iK}, \tilde{y}_{iK}), x_{iq}]$. We seek to score each head by its causal indirect effect toward predicting the correct $y_{iq}$ from the shuffled prompt $\tilde{p}_i^t$; that is, attention heads that promote task accuracy in the context of the shuffled prompt. To compute this, denote $a_{lj} := \bar{a}_{lj}^t$ the intervention of setting the output of attention head $a_{lj}$ to its task-conditioned activity $\bar{a}_{lj}^t$. Then compute the difference in probabilities assigned to the first token of the correct $y_{iq}$ between when we intervene on the model $f$ and when we do not: $CIE(a_{lj} \mid \tilde{p}_i^t) = f(\tilde{p}_i^t \mid a_{lj} := \bar{a}_{lj}^t)[y_{iq}] - f(\tilde{p}_i^t)[y_{iq}]$. Todd et al. (2024) find that a small set of heads consistently achieve high causal scores across their task set. Using $\mathcal{A}^D$ to denote this set of top heads, FVs are computed for each task using the heads' task-conditioned means: $v_t = \sum_{a_{lj} \in \mathcal{A}^D} \bar{a}_{lj}^t$.

### 2.1 Generalizing function vectors beyond in-context demonstrations

One of our contributions is to generalize the function vector method described above from demonstrations to alternative task specifications. We denote by $Q_t$ a set of task specifications (of any form) for the task $t$ (e.g., the instructions in Figure 1). For a query example $(x_{iq}, y_{iq})$ (Egypt $\Rightarrow$ Cairo in Figure 1), we sample $q_m^t \in Q_t$ and construct prompts $p_i^t$ as $p_i^t = [q_m^t, x_{iq}]$ Next, we address the challenge of creating uninformative baselines $\tilde{q}_m^t$ for the task specifications $q_m^t$. We consider three approaches to generating these baselines, and see further details and examples in Appendix B:

- **Equiprobable token sequences.** *Intuition:* sample token sequences that are similarly likely under the model but are unrelated to the task. Each $q_m^t$ is encoded as a token sequence $q_m^t = [w_1, w_2, ..., w_L]$ whose probability under the model $f$ is $P(q_m^t) = \prod_{l \leq L} P(w_l \mid w_{<l}) = f(w_{<l})[w_l]$. Starting from the BOS token, we sample $\tilde{w}_l$ to approximately match the conditional probabilities $f(\tilde{w}_{<l})[\tilde{w}_l] \approx f(w_{<l})[w_l]$, and construct $\tilde{q}_m^t = [\tilde{w}_1, \tilde{w}_2, \cdots, \tilde{w}_L]$.

- **Real texts:** *Intuition:* sample texts from a natural corpus that convey no task information but otherwise match the task specifications. We score token sequences from a chosen corpus under the model and sample $\tilde{q}_m^t$ with approximately the same length and probability as $q_m^t$.

- **Other task specifications:** *Intuition:* sample task specifications for other tasks $t' \neq t$ that likely convey no information for task $t$. We score task specifications $q_{m'}^{t'}$ generated for other tasks $t' \in \mathcal{T}$, and again sample ones with approximately the same length and probability as $q_m^t$.

Following Todd et al.'s (2024) selection of prompts in which the model successfully performs the demonstrated task, we focus on a best-performing small set of alternative task specifications we denote as $Q_t^* = \{q_1^{t*}, \cdots, q_J^{t*}\} \in Q_t$, with the intuition that these facilitate forming the most salient representation of the task $t$. We construct prompts $p_i^t = [q_m^t, x_{ik}]$, keep only ones in which the model successfully predicts $y_{ik}$, use these to select $Q_t^*$, and narrow down our set of prompts to a final $P_t^*$, where $p_i^{t*} = [q_m^{t*}, x_{ik}]$. From there we follow the original function vector procedure: we compute task-conditioned activations (over $P_t^*$), generate uninformative baselines $\tilde{p}_i^{t*} = [\tilde{q}_m^{t*}, x_{ik}]$, compute the indirect effects for each attention head, and use these to select the top heads (in practice, we average over these different baselines in computing head causal effects; see below).

**Constructing function vectors from in-context instructions** Although the method is more general, here we use zero-shot textual instructions as our $Q_t$. We generate a candidate set of textual

instructions for each task by querying Llama-3.1-405B (Llama Team, 2024) using a $K$-shot ICL prompt for the task, and ask it to generate 10 instruction prompts for the task it infers from the provided demonstrations (see Appendix A for the templates). We repeat this procedure 20 times for each task $t$ and deduplicate exact repetitions to arrive at a task instruction set $Q_t$. We repeat this procedure twice, once encouraging the model to generate short instructions and once with no such encouragement. We provide sample generated instructions of both lengths in Appendix A.1 and corresponding uninformative baselines (using all three approaches) in Appendix B.2. For our 'real texts' uninformative baseline, we sample texts from WikiText-103-v1 (Merity et al., 2016).

## 2.2 Experimental conditions

**Hyperparameters.** We match the settings used by Todd et al. (2024): we use 100 examples to compute mean task-conditioned activations and 25 for the indirect effects. We do so over the $J = 5$ top textual instructions for each task, splitting the examples evenly between the top instructions.

**Models.** We focus on the base and instruction-tuned versions of the 3B Llama-3.2 and 8B Llama-3.1 models, with the full list in Table 4. We also report some results with the weaker 1B Llama-3.2 and 7B Llama-2 models (Touvron et al., 2023), the latter of which matches Todd et al.'s (2024). Finally, to examine the roles of post-training stages, we evaluate four OLMo-2 models (OLMo et al., 2024).

**Tasks.** We consider the same set of tasks and datasets used by Todd et al. (2024). We omit a few classification datasets where successfully predicting the next token requires an understanding of the format that is facilitated by demonstrations but not necessarily by minimal textual instructions (we retain a total of 50 datasets; see Appendix D for the full list). We follow Todd et al. (2024) in computing the top sets of heads only over datasets where a model surpasses chance performance. As our approach requires 20 successful prompts for each of the five best instructions to compute the mean task-conditioned activations, we also omit tasks where a model fails to pass this number of prompts (which, in most cases, means it is also below chance accuracy and was already omitted ).

**Textual instructions and uninformative baselines.** We evaluate each model on each task using one random seed six separate times, for the short ($\leq 16$ tokens) vs. longer (unbounded) instructions crossed with all three uninformative baseline approaches (§2.1). We observe minimal deviation in top heads across these conditions (see Appendix H.3), so we average the causal effects across all six to compute the top heads. We then report final evaluation accuracies averaged over the results with both sets of mean activations, those generated with short instructions and those from longer ones.

**Results using in-context demonstrations.** To enable comparison, we also replicate the original, demonstration-based function vector evaluation with all models we consider, using the same random seeds (and therefore, same train-evaluation splits) as used in the textual instruction setting.

**Evaluation and Comparison Logic.** We structure our evaluations to facilitate comparison between FVs derived from demonstrations and instructions. We use the $|\mathcal{A}| = 20$ top heads in both settings, consistent with Todd et al. (2024) for the 7B models. We follow Todd et al. (2024) in evaluating function vectors as an additive intervention to the residual stream after the $\lfloor L/3 \rfloor$ layer (layer 9 for the Llama-3.2-3B models and layer 11 for the Llama-3.1-8B and OLMo-2-1123-7B ones). We report two evaluation settings: 10-shot with shuffled labels ($\tilde{p}_i^t$ in §2.1) and 0-shot (with no instructions). We focus our assessment of each FV in the setting matching its extraction: shuffled 10-shot for demonstration FVs and 0-shot for instruction FVs. In addition to the FV evaluations, we report a baseline of the evaluation setting without adding the function vector. We report accuracies using informative task presentations for each evaluation in Appendix F: instructed 0-shot (averaging over the top $J = 5$ instructions for each model and task), and 10-shot (without label shuffling).

## 3 Results

### 3.1 Function vectors extracted from instructions facilitate zero-shot task accuracy

Figure 2A summarizes our evaluation of demonstration function vectors (left) and instruction-based FVs (right). We find that **our adaptation of the FV extraction procedure succeeds and increases zero-shot task accuracy** from below 20% to above 50% in the best models. A notable exception is the Llama-3.1-8B model, for which the $\lfloor L/3 \rfloor$ intervention depth appears suboptimal (see results from other intervention layers in Figure 11). On the other hand, the Llama-3.2 models may benefit

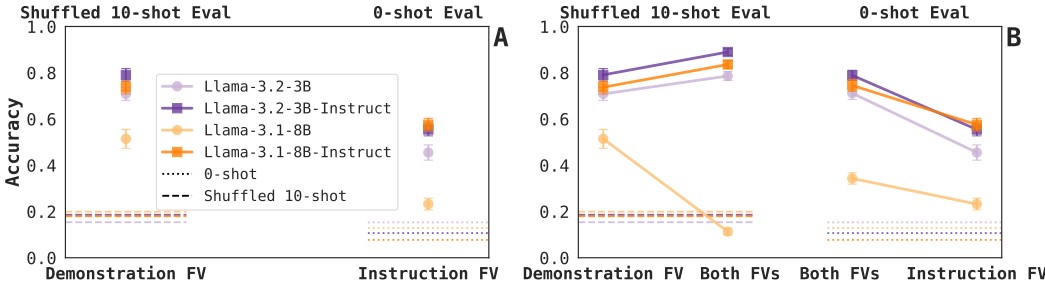

**Figure 2: Instruction-based function vectors are effective in zero-shot evaluation; both function vectors are beneficial together. (A)** We evaluate each FV in the setting matching its extraction: shuffled 10-shot for demonstrations and 0-shot for instructions. Both FV types are effective at their respective evaluations, though demonstration ones fare better. This procedure is effective for both post-trained models and for the distilled Llama-3.2-3B base model; less so for the Llama-3.1-8B base model. **(B)** We examine the effect of jointly adding both FVs together. The joint intervention outperforms either one by itself (with the exception of the base Llama-3.1-8B model, for which the $|L/3|$ intervention depth appears highly suboptimal; see Figure 11). Dotted lines represent baselines with no FV intervention and error bars reflect standard errors of the mean (SEMs). See Figure 11 for results when intervening at optimal depths, rather than a fixed $|L/3|$ depth, and Figure 12 for results evaluating the function vectors in the setting opposite to their extraction.

from being distilled from larger, post-trained Llama-3.1 models. The top $|\mathcal{A}| = 20$ attention heads also represent a slightly larger fraction of attention heads in the smaller models (about $3\%$ in the 3B models vs. $2\%$ in the 8B ones). We also observe that, unsurprisingly, instruction FVs are substantially more effective in post-trained models. Finally, while instruction FVs are highly effective in the zero-shot setting, they fail to match the accuracy demonstration FVs attain in the shuffled 10-shot condition, and struggle when evaluated in the shuffled 10-shot condition themselves (see Figure 12). We replicate findings 1 and 2 (below) with other models: see Figures 13-15, in appendices G.3-G.4.

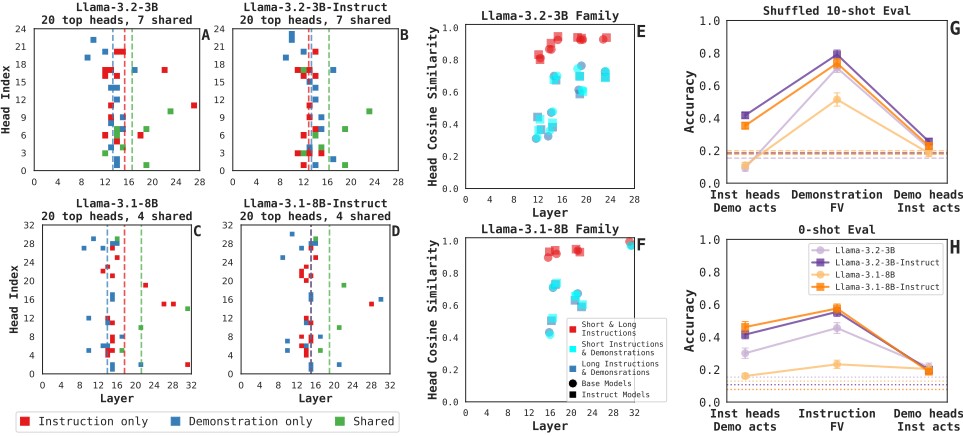

**Figure 3: (A-D) Demonstrations and instructions elicit mostly distinct attention heads.** We visualize the sets of top heads identified only with demonstrations, only with instructions, and shared in both cases. Only a few top heads are shared each in model (see Appendix H.1 for additional models). We observe that top instruction FV heads tend to be in earlier layers of the post-trained (-Instruct) models, compared to the base versions, and that shared heads tend to arise in later layers. **(E-F) Shared head activation similarity depends on layer, but not on post-training.** We plot the cosine similarity of activations in shared heads between three sets of task-conditioned activations: those elicited by demonstrations, by short instructions, and by longer instructions. We observe increasing similarity between demonstrations and instructions in later model layers, but show no effect of post-training on similarity. **(G-H) 'Heterogeneous' function vectors show demonstration-instruction asymmetry.** We construct 'heterogeneous' FVs using attention heads localized with instructions and demonstration task activations (left data points) to the opposite combination, using demonstration-localized heads and instruction task activations (right). Compared to the 'homogeneous' function vectors (middle), we mostly observe a smaller accuracy drop when using instruction FV heads to read from demonstration mean activations than vice versa.

We attribute these differences to two factors. Demonstration FVs arise from the attention heads that most improve shuffled 10-shot accuracy (as those were the uninformative prompts $\tilde{p}_i^t$; §2.1). In contrast, 0-shot evaluation differs from the instruction FV baseline prompts, as it contains no instructions (and hence, has fewer tokens than the uninformative instructions $\tilde{q}_t^i$; §2.2). Further, ICL from demonstrations appears more primal to LLMs than from instruction-following. The former appears in base models, while the latter requires post-training; we support this claim with later results.

Constructing function vectors seems to benefit from model capabilities. Smaller Llama-3.2-1B models show a lower accuracy increase with instruction FVs; conversely, newer OLMo-2-1124-7B models show a greater increase (see Appendix G.4). This capacity is not unique to the latest models; results with Llama-2-7B models qualitatively match newer models. Finally, we evaluate the effect of choosing a fixed intervention layer in Figure 11. If we choose the optimal layer for each model and evaluation, the accuracy numbers rise as expected, but the overall patterns remain the same.

## 3.2 Demonstration and instruction function vectors are beneficial together

If demonstrations and instructions elicit different task representations in a language model, to what extent are their representations jointly beneficial? To study this, we intervene with both FVs, adding them to the residual stream after the same $\lfloor L/3 \rfloor$ layer. Our results in Figure 2B demonstrate that **adding both function vectors appears consistently beneficial**, except for the base Llama-3.1-8B model (for which the choice of layer seems crucial). Surprisingly, adding both FVs after the same layer does not appear to induce interference, even though these FVs were extracted independently. In Figure 11B, we report results from the highest accuracy intervention layers in a sweep over layer pairs; adding both FVs to the same layer often proved optimal. Finally, to establish how much of this effect is simply due to amplifying the intervention, we report the results of adding the same FV twice in Figure 18. For some models and evaluations, adding the same FV twice performs similarly to adding both FVs. This suggests that some of the benefit is due to the magnitude of the intervention rather than the different information carried in both FVs; we leave studying the extent of these relative effects to future work. Having found behavioral evidence that these different function vectors convey different information, we next examine to what extent they share mechanisms within the model.

## 3.3 Instruction and demonstration FVs mostly engage different attention heads

In Figure 3A-D, we visualize the $|\mathcal{A}| = 20$ top attention heads identified in each model by the FV extraction procedure. We observe that **demonstrations and instructions elicit mostly distinct sets of heads.** Of the 20 top heads identified, instruction and demonstration FVs share few of them – 7 in both Llama-3.2-3B models and only 4 in the Llama-3-1.8B ones. Post-training appears to move instruction FV heads closer to demonstration FV heads. The most notable change between base and post-trained model versions is in the average layer of the instruction-only heads — from a mean several layers deeper than demonstrations in base models, to almost identical depths in post-trained ones (other model classes are similar, see Appendix H). We observe higher causal scores in demonstration top heads. In §2, we define the causal score of each head as its contribution to correctly predicting the next token. Demonstration FV top heads receive substantially higher scores than instruction ones (Table 7). This suggests that task inference from demonstrations is more localized to a small set of heads than from instructions and that instruction-based task inference is more diffuse.

We also examine the mean task-conditioned activation patterns in the top heads shared between instructions and demonstrations. We compute the cosine similarities between the patterns elicited by demonstrations vs. by shorter/longer instructions (see Figure 3E-F). Expectedly, we find higher similarities between the two instruction-driven activation patterns than between the instruction and demonstration patterns. Additionally, later layers are generally more similar. Finally, post-training appears *not* to adapt demonstration FV heads. We observe no difference in demonstration-instruction similarities between base and post-trained models, and equal numbers of shared heads between instructions and demonstrations across model versions. Instead, post-training appears to induce a different, separate mechanism for instruction task inference, one we hope to explore in future work.

### 3.4 Instruction-identified attention heads are more useful for building demonstration FVs than vice versa

If demonstrations and instructions share a common task inference mechanism, we would not observe a significant difference if we constructed 'incongruent' function vectors using top heads identified by one with activations from the other. We borrow from neuroimaging and consider the FV extraction procedure through the lens of functional localizers, viewing head identification as localization and mean activation computation as recording (Saxe et al., 2006; Berman et al., 2010). We evaluate both types of 'incongruent' FVs, those constructed with demonstration top heads and those using instruction top heads, and summarize our results in Figure 3G-H. We observe that, as expected, **accuracy with these incongruent FVs falls below the regular FVs in all cases**. We note an asymmetry: **using instruction-localized top heads from with demonstration mean activations seems preferable to the opposite combination** (with the exception of the shuffled 10-shot evaluation on the base models). This effect is sharpened when we evaluate each FV in the layer resulting in the highest accuracy, rather than fixing to $|L/3|$ (Figure 28). To explain this, we report the causal indirect effects of the top heads from each task presentation in the opposite presentation format – that is, the scores of instruction-localized heads when using demonstrations, and vice versa (Table 8). Examining the scores, we observe that instruction-localized heads are more helpful in the demonstration setting than demonstration-localized heads are in the instruction setting. We take this as evidence that the mechanism for task inference from instruction leverages attention heads that play a minor role in demonstration task inference. The development of such a mechanism may also explain the depth alignment shifts we previously observed between base and post-trained models (subsection 3.3, Figure 3A-D) In contrast, the demonstration task inference mechanism uses attention heads that are less useful for instruction task inference. This supports our earlier claim about the primacy of demonstration ICL. We conclude with two control experiments to ensure that this observed asymmetry is meaningful (and not an artifact of selecting arbitrary sets of heads). We select sets of heads that are either unrelated to both FV types, or that have the *lowest* causal scores (Appendix J.1) In both cases, accuracy falls to task baseline or below, suggesting the observed effect is meaningful.

### 3.5 Instruction FVs from post-trained can steer base models, arise from SFT and DPO

Given the effectiveness of instruction FVs in post-trained models, we examine whether instruction FVs from post-trained models can steer base models. We repeat the instruction FV evaluations, intervening on each base model with the FV generated by its post-trained version (multiple post-trained variants exist for OLMo-2-1124-7B; we use the final model, OLMo-2-1124-7B-Instruct). We report our results in Figure 4B. In three of the base models, **we find substantial accuracy increases, nearing the zero-shot accuracy elicited by instruction FVs in the post-trained models** (consistent with Stolfo et al.'s (2024) cross-model steering results). This effect is even more striking when

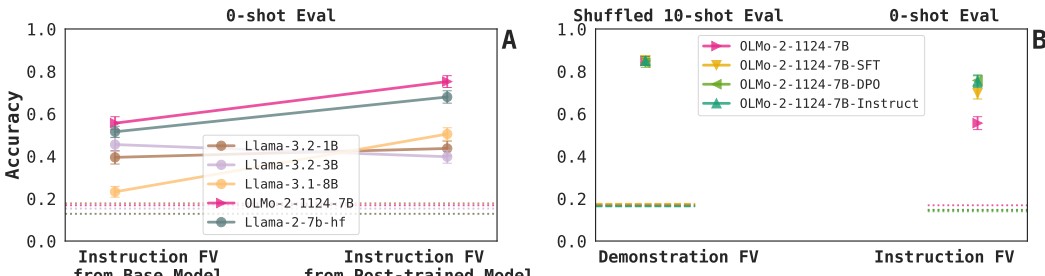

**Figure 4: Instruction-based function vectors can transfer from post-trained to base models and are facilitated by SFT and DPO. (A)** We apply the instruction FVs extracted from post-trained models to their respective base models. We find weak impacts on the distilled Llama-3.2 models, and substantial benefits in the other ones, almost recovering the post-trained model FV evaluation accuracy. **(B)** We examine the OLMo-2-1124-7B family of models and observe meaningful increases in the efficacy of instruction-based function vectors in both the SFT and DPO stages over the base model; conversely, the final RL stage appears minimally impactful. In both panels, dotted lines represent baselines with no FV intervention, and error bars reflect standard errors of the mean (SEMs). See Figure 33 for results evaluating the function vectors in the setting opposite to their extraction, and Figure 34 for results when intervening at optimal depths, rather than a fixed $|L/3|$ depth.

evaluating the steering instruction FVs in the shuffled 10-shot setting (Figure 33A). We take this as further evidence that this procedure identifies the task representations elicited by instructions. We find minimal or mildly negative effects on the Llama-3.2 models.

We conclude by examining which post-training stages contribute to the ability to extract instruction FVs using the OLMo-2-1124-7B family of models (OLMo et al., 2024). The authors release four versions of the model: a base version, one following supervised fine-tuning (-SFT), another following preference fine-tuning (-DPO), and a final version (-Instruct). Figure 4B depicts results with these four models. All OLMo models are conducive to extracting function vectors from demonstrations, but the base model instruction FVs show lower accuracies. We observe two accuracy increases, once with the SFT model and another with the DPO model. The DPO model also offers an increase in the efficacy of instruction FVs in mismatched evaluations (see Figure 33B). Finally, we note that post-training appears to have no impact on demonstration-based ICL, as evidenced by the lack of variability in Figure 4B (left).

## 4 Related Work

**In-Context Learning.** The ability to perform tasks from demonstrations has been in the foreground of language model research following its identification by Brown et al. (2020). Substantial subsequent research has studied LLMs' ability to perform in-context learning, from perspectives such as the data distribution (Chan et al., 2022; Chen et al., 2024), learning algorithms (Xie et al., 2021; Akyürek et al., 2022, 2024), mechanics and learning dynamics (Von Oswald et al., 2023; Zhang et al., 2023; Park et al., 2024; Han et al., 2024; Li et al., 2025), and latent representations (Pan et al., 2023; Todd et al., 2024; Hendel et al., 2023; Yin and Steinhardt, 2025). Other lines of work examined factors to ICL contributing (Min et al., 2022), such as sensitivity to in-context prompt formatting choices (Sclar et al., 2023; Su et al., 2025) Most prior work focuses on ICL from demonstrations; our work bridges between those and instructions as a gateway to exploring other forms of ICL (Lampinen et al., 2024).

**Instruction-following** refers to executing a task given a natural language, similar to how one person may instruct another. Although fine-tuning is necessary to elicit this capability, it has been shown to facilitate generalization and usefulness following the introduction of the Flan models (Chung et al., 2022). Instruction-following has since been the focus of dataset development (Mishra et al., 2022; Wang et al., 2022a,b; Zhou et al., 2023a; Taori et al., 2023), model evaluation (Zhou et al., 2023b; Liu et al., 2024; Lyu et al., 2024), and interpretability work (Stolfo et al., 2024; Wu et al., 2024). Our work explores whether instruction tuning leverages the mechanisms of demonstration-based ICL.

**Task Representations.** Computational neuroscience has long sought to understand how neural networks learn to represent abstract tasks (Cohen et al., 1990; Botvinick and Plaut, 2002; Yang et al., 2019; Flesch et al., 2021; Farrell et al., 2023; Hummos et al., 2024). In tandem, computer scientists devised methods to explicitly introduce these representations to models (Lampinen and McClelland, 2020; Ilharco et al., 2022; Shao et al., 2022). Our work follows a recent line of inquiry extracting task representations that arise in LLMs (Todd et al., 2024; Hendel et al., 2023; Saglam et al., 2025) or VLMs (Luo et al., 2024; Hojel et al., 2024; Huang et al., 2024) as they complete tasks. Most task representation extraction approaches fall into one of a few categories. Some treat the model's latent (residual) representation of a delimiter token at some depth as the task representation (e.g., Hendel et al., 2023; Luo et al., 2024; Li et al., 2025). Another approach learns a per-layer task representation (Saglam et al., 2025). Finally, Todd et al. (2024) combine activity from multiple attention heads across different depths and intervene with the resultant function vector at a single depth. We focus on contrasting the representations that arise from different textual presentations of the same task.

## 5 Discussion

We empirically test for the presence of common, presentation-agnostic task representations in large language models. Our results suggest that different presentations of a task *do not* elicit a common task representation through function vectors. We extend the FV extraction procedure from in-context demonstrations to arbitrary task presentations and successfully construct instruction FVs that promote zero-shot accuracy. We find these convey different information than demonstration FVs, evidenced by the benefits of intervening with both forms of function vectors together over either by itself. We also find that mostly distinct (though partly overlapping) sets of attention heads causally mediate task

performance in these two settings. Our results offer support (in a limited, but controlled setting) for the widely used practice of prompting language models with both demonstrations and instructions. Our findings support two other takeaways. First, ICL from demonstrations may be more inherent to LLMs. We offer preliminary evidence that instruction task inference leverages attention heads that are peripherally useful for demonstration-based ICL. The contrary effect is weaker; demonstration FV attention heads are less helpful for instructions. Second, we show evidence that instruction FVs transfer from post-trained models to steer their base versions better than base model instruction FVs.

We found the effectiveness of instruction-derived FVs surprising. Given the diversity of plausible instructions for a given task, we would have considered it a priori likely that instructions induced relatively diffuse representations that would not support FV extraction from a few attention heads. Indeed, the representation of textual instructions appears more diffuse in language models, as evident from the distributions of attention head causal scores (Table 7); yet, our approach is successful.

The accuracy discrepancy when function vectors are evaluated in a setting incongruent with their extraction further supports the notion that task representations are dependent on the form of presentation (compare Figure 2 and Figure 12). This evaluation form-dependency portends a challenge in identifying, monitoring, or amplifying task representations in naturalistic settings, such as prompts combining multiple demonstration forms, obfuscated or adversarially-presented tasks, or other more benign presentations forms, such as role prompts or explanations (Lampinen et al., 2024).

In our results, we average over the three uninformative baselines we propose: sampling tokens directly from the model, natural texts, and instructions from other tasks. While the selected top heads did not greatly differ between the three, one is unlike the other two (Appendix H.3). Instructions from other tasks still convey the existence of a task, just a different one, and may help localize task-specific information. Conversely, tokens sampled from the model and natural texts do not convey 'task-ness,' hence helping to localize the existence of a task in general. The differences between attention heads identified with each baseline alone may elucidate the differing roles of instruction FV attention heads.

We identify which post-training stages contribute to the ability to extract instruction FVs. However, our results do not explain *what* takes place in post-training that facilitates inferring tasks from instructions. Wu et al. (2024) propose that the primary drivers could be changes in processing words that convey instructions—which suggests the hypothesis that the instruction FV attention heads should strongly respond to these tokens. In another recent work, Yin and Steinhardt (2025) compare induction heads (Elhage et al., 2021) and FV heads—do instruction FV heads share anything in common with induction heads as well? On that note, what can we learn from the heads that change in importance between base and post-trained models? Our results point a spotlight at a set of heads relevant to this capacity; future work may study their contributions and mechanisms in greater depth.

Our finding that we can steer base models with post-trained models instruction FV is consistent with recent findings by Stolfo et al. (2024) on activation steering in an instruction-following context. Our current results do not suffice to offer a potential savings in the effort required to post-train a model, as so far, we only demonstrate the ability to steer a base model with its own post-trained variation. Recent results by Lee et al. (2025) propose a method for transferring steering vectors between models — combined with our results, this suggests a potential to confer some of the benefits of post-training on a model that has not been post-trained, a connection we hope to explore in future work.

Finally, we highlight a few other proposals on how our work might inform prompt and intervention design. Our finding regarding the diffuseness of instruction FVs (subsection 3.3) implies that interventions focused on changing or improving instruction-following may benefit from intervening at multiple locations and layers; presently, most steering methods perform a single additive residual stream intervention, which may be more meaningfully suboptimal for instructions, than for example, for demonstrations. The observation that demonstrations and instructions elicit different FV attention heads (also in subsection 3.3) suggests a speculative potential pathway to improving prompt design. In contexts where both instructions and demonstrations are provided, it may be possible to monitor their separate sets of implicated attention heads, and try to classify from their activity whether or not the model successfully formed a task representation. This classifier could serve as a signal that helps guide toward improving the instructions or selecting different or additional demonstrations.

## 5.1 Limitations

**Task set.** We use a limited set of minimal, fairly simple tasks, following Todd et al.'s (2024) choices. We omit a few classification tasks where predicting the correct token requires understanding the expected answer format, information conveyed clearly by demonstrations, but not necessarily by instructions. This could be addressed by combining demonstrations and instructions, or by encouraging the instruction-generating model to include output formatting information. Furthermore, it would be of interest to examine how these findings generalize to longer, more complex, open-ended, and naturalistic tasks, such as the ones present in MMLU (Hendrycks et al., 2020) or BBH (Srivastava et al., 2022; Suzgun et al., 2022).

**Choice of task representation.** We study a particular proposal for task representations — function vectors as formulated by Todd et al. (2024). Other task representation extraction approaches exist in the literature, such as task vectors (Hendel et al., 2023) — we hope future work examines whether our findings translate to their formulation as well. More broadly, we cannot conclusively prove the negative, that common task representations do not exist; we can only demonstrate that our methods did not identify them. We also do not explore the working of attention heads identified in greater depth, such as identifying which circuits (Elhage et al., 2021) they may contribute to.

**Arbitrary intervention depth.** We follow Todd et al. (2024) in intervening at a fixed $|L/3|$ depth. While we also present (in the appendices) results at empirically optimal intervention depths for each model, the choice retains a measure of arbitrariness. It is possible, for instance that different types of tasks are better served better by different intervention depths, and our analyses do not reflect that.

**Model sizes.** We focus our investigation on smaller (large) language models. While we offer fairly consistent evidence across the models we examine, we do not explore scaling with model size. We note that Todd et al. (2024) successfully extracted demonstration function vectors to Llama-2 models at the 7B, 13B, and 70B sizes, which offers promising evidence for scaling function vectors.

**Limited replications.** We evaluate many experimental conditions (Appendix E.1). For each model we study, we evaluate each of the approximately 50 tasks six times — shorter and longer instructions, each with each of the three baselines. We also ran the additional control conditions reported with all tasks for the Llama-3.2-3B and Llama-3.1-8B models. However, we only replicate each model on each dataset in each setting once—given the qualitatively consistent results observed by Todd et al. (2024), replicating over random seeds (used primarily in train-test splits and query example sampling) seemed untenable, if not wasteful. We use a single prompt template (Figure 9), relying on consistent previous results with different prompt templates (p. 6 and Appendix C in Todd et al., 2024).

## Acknowledgments

We thank members of the Computation and Cognition lab at NYU and the Human and Machine Intelligence lab at Princeton for valuable feedback at various stages of this work. We thank Eric Todd, Eshika Saxena, and Nicola Cancedda for feedback on this project and manuscript. Author GD was funded by the Meta AI Mentorship program during his work on this manuscript.

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

# A Prompt generation prompt templates

```
<|start_header_id|>user<|end_header_id|>
```
**Instructions**
You are powerful model helping write prompts to help smaller models perform tasks better. Below, you will be given a set of input-output pairs for a particular undescribed task. First, please study the examples to deduce what the task is, and describe your thinking under the header "# Task Deduction". Next, please write 10 prompts that might help a smaller model perform this task. The prompts should be:

1. Short, up to 10 words.

2. Informative about what the task is.

3. Not repetitive with each other.

Please write your prompts under the header "# Task Prompts".

**Task examples**
{task examples}

Now, think step by step and follow the instructions above.
```
<|eot_id|><|start_header_id|>assistant<|end_header_id|>
```

**Figure 5: Prompt for generating short task instructions given ICL examples.** The prompted model (in our experiments, Llama-3.1-405B) must analyze the examples given under the header "task examples" and generate instructions for another model. We provide no information beyond the in-context examples, and repeat this procedure 20 times to generate a set of approximately 200 instructions for each task, which we then deduplicate. In practice, the vast majority of these short instructions tokenized to 16 or fewer tokens, so we set that as the limit for what we considered a short instruction.

```
<|start_header_id|>user<|end_header_id|>
```
**Instructions**
You are powerful model helping write prompts to help smaller models perform tasks better. Below, you will be given a set of input-output pairs for a particular undescribed task. First, please study the examples to deduce what the task is, and describe your thinking under the header "# Task Deduction". Next, please write 10 prompts that might help a smaller model perform this task. The prompts should be:

1. As long as necessary to be helpful for the smaller model.

2. Informative about what the task is.

3. Not repetitive with each other.

4. Not including any examples of the task.

Please write your prompts under the header "# Task Prompts".

**Task examples**
{task examples}

Now, think step by step and follow the instructions above.
```
<|eot_id|><|start_header_id|>assistant<|end_header_id|>
```

**Figure 6: Prompt for generating longer task instructions given ICL examples.** The prompted model (in our experiments, Llama-3.1-405B) must analyze the examples given under the header "task examples" and generate instructions for another model. We provide no information beyond the in-context examples, and repeat this procedure 20 times to generate a set of approximately 200 instructions for each task, which we then deduplicate. In practice, the vast majority of these longer instructions tokenized to 64 or fewer tokens, so we set that as the limit for what we considered a long instruction.

## A.1 Example generated instruction prompts

**Table 2: Example short instructions for five tasks.**

| Task | Short Instructions |
|------|--------------------|
| **antonym** | Create an opposing term
Identify the antithesis of this word
Create a counter-term
Reverse the semantic meaning
Provide a word that is the semantic opposite |
| **country-capital** | Country to capital city correlation
Learn country-capital associations
Map country names to their capitals
Identify the administrative center
Provide the capital city for the given country |
| **concept_v_object_5** | Select the word that is not a noun
"Find the word that is not a concrete object."
Select the word that tells us more about something
Which word has a distinct semantic meaning?
Identify the adverb or adjective in the list |
| **english-spanish** | Spanish equivalent for this English term
Translate everyday English words to Spanish
Spanish translation of English word
Find Spanish counterpart for English word
Find Spanish translation |
| **product-company** | "Associate product name with company name"
Which company created this software?
"Classify product by owner company"
Identify the company that developed this technology
"Link this device to its manufacturer." |

**Table 3: Example long instructions for five tasks.**

| Task | Long Instructions |
|------|-------------------|
| **antonym** | Find a word that, when compared to the input word, presents a contrasting meaning. This word should highlight the differences and serve as an antonym
Generate a word that cancels out the meaning of the input word
**Meaning reversal**: Reverse the meaning of the input word by generating a word that represents its opposite. Ensure that the generated word is semantically accurate and contextually relevant
This task tests the ability to navigate the vocabulary of a language to find and generate antonyms. Please focus on producing words that are directly opposite or clearly contrasting
**Find a word that contrasts with the input word in meaning.** This could involve finding a word that is the opposite of the input word or one that describes a different extreme or end of a spectrum |
| **country-capital** | What is the name of the city where a country's president or monarch typically resides and conducts official business?
Determine the capital city of a country by identifying the city where the national government is seated and where major political decisions are made
Provide the name of the city that is generally accepted as the capital of a particular country
What city is recognized as the center of administration and governance for a given country?
"Countries around the world each have a capital city where their government is based. Your task is to know what these cities are for any country you are asked about." |
| **concept_v_object_5** | Determine the word in the list that is a verb or an action
Identify the word in the list that describes a quality, property, or characteristic of something
Identify the word in the list that describes a quality or property of something
**Determine the Quality Word**: Determine which word from the list describes a quality, state, or condition. This word should tell us about the nature or attributes of something
Find the word that can be used in a sentence to describe an action, event, or situation |
| **english-spanish** | Translate the English word into Spanish, making sure to use the most appropriate and commonly used term in Spanish-speaking contexts
Provide a Spanish translation of the input word that is both accurate and fluent
Translate the input word from English to Spanish, considering any relevant context or connotations
Translate the given English word into its equivalent in Spanish, ensuring to maintain the original meaning and word type (noun, verb, adjective, etc.)
Identify the Spanish equivalent of the provided English term, ensuring the translation is accurate and suitable for the context |
| **product-company** | Identify the developer of a given operating system, platform, or tool
Given the name of a product, technology, or format, find the company that owns or developed it. Use your knowledge of industry leaders and their offerings
Identify the company or organization that developed or owns the product, technology, or format specified in the input
Identify the company that created this file format
Determine the company that is associated with the specified brand, product, or format |

# B  Uninformative instruction baselines

## B.1  Additional uninformative instruction details

**Equiprobable token sequences.**  Given task-informative instructions $q_m^t$, our goal is to sample an uninformative instruction $\tilde{q}_m^t$. We encode $q_m^t$ as a token sequence $q_m^t = [w_1, w_2, ..., w_L]$ whose probability under the model $f$ is $P(q_m^t) = \prod_{l \leq L} P(w_l \mid w_{<l}) = f(w_{<l})[w_l]$. We begin from the BOS token, and sequentially sample $\tilde{w}_l$ to create $\tilde{q}_m^t = [\tilde{w}_1, \tilde{w}_2, \cdots, \tilde{w}_L]$. We do so by approximately matching the conditional probabilities $f(\tilde{w}_{<l})[\tilde{w}_l] \approx f(w_{<l})[w_l]$ using the following logic. First, we mask out any non-text ("added vocabulary") tokens. At each step, we compute the (log-) probability of the $l$'th token, $\log P(w_l \mid w_{<l})$. We compute the distribution over next tokens of the uninformative instructions, $\log P(\cdot \mid \tilde{w}_{<l})$. We set an initial threshold and a threshold increment (in our experiments, both were set to 0.1). We increase the threshold by the increment until at least one token in $\log P(\cdot \mid \tilde{w}_{<l})$ falls within the incremented threshold of the log-probability of the current instruction token. Denote by $t$ our initial threshold, $\Delta t$ our increment, and $k \in \mathbb{Z}^*$ the number of increments required, we find the following: $\min_k : \left( \sum_{\tilde{w}_l'} \mathbb{1}[t - k\Delta t \leq \log |P(w_l \mid w_{<l}) - \log P(\tilde{w}_l' \mid \tilde{w}_{<l})| \leq t + k\Delta t] \right) > 0$, where $\mathbb{1}[\cdot]$ denotes the indicator function. We then sample uniformly between all $\tilde{w}_l'$ satisfying the previous relation and append the sampled $\tilde{w}_l$ to the $\tilde{q}_m^t$ we are constructing, continuing until we reach the same length as the informative $q_m^t$.

**Real texts.**  Ahead of time, we precompute the log-probability of texts from WikiText-103-v1 (Merity et al., 2016) with each model $f$. We randomly sample entries from the WikiText-103-v1 dataset. For each sampled entry, we extract overlapping prefixes that end with whitespace (excluding the terminal whitespaces), tokenize these prefixes, and keep those with a length of 64 or fewer tokens. We extract prefixes to create strings that might coherently appear at the beginning of a text. We then compute and cache the (log-) probability of each such prefix and its token sequence. We cache approximately $2^{16} = 65536$ token sequences with each model (approximately as we stop after the Wikitext entry that brought us over the threshold, but do not discard sequences beyond the $2^{16}$'th one).

To sample an uninformative $\tilde{q}_m^t$ for some $q_m^t$, we tokenize $q_m^t$ to arrive at its length $L(q_m^t)$ and compute its log-probability $\log f(q_m^t)$. We now create a representative sample of $N = 100$ cached texts with approximately the same length as $L(q_m^t)$. We begin by only consider texts of with a length of precisely $L(q_m^t)$ tokens. If there are over $N$ of those, we stop; otherwise, for $k \in \mathbb{Z}^+$, we also consider texts of length $L(q_m^t) \pm k$, increasing $k$ by one until we have a set of at least $N$ candidates. Once we have attained this set of candidates, we select the $\tilde{q}_m^t$ with the closest log-probability under $f$ as $q_m^t$ has, and remove it from the set. As we require five uninformative baselines for each instruction $q_m^t$ (see Appendix E), we will use the five real texts with the closest log-probability to $q_m^t$ subject to being within a small number of tokens from $L(q_m^t)$.

**Other task instructions.**  We follow a conceptually similar procedure to the one described above for real texts, but using instructions generated for other tasks. As this is a smaller set, we do not cache these log probabilities in advance. We exclude instructions generated for the same task, but otherwise follow an identical procedure. Denote by $L(q_m^t)$ the length of the instructions $q_m^t$. We identify a set of candidate alternative instructions with a length of approximately $L(q_m^t)$, increasing the acceptable difference in length until we reach $N = 100$ candidates. We then return the one with the closest log-probability to $q_m^t$ under the model and remove it from the set, so as before, we will use the five other task instructions with the closest log-probability to $q_m^t$ subject to being within a small number of tokens from $L(q_m^t)$.

## B.2  Prompt baseline examples

| Task | Equiprobable tokens | Real texts | Other task instructions |
|---|---|---|---|
| **antonym**
*Create an opposing term* | - Success bearings line example
- ilocราชงานกระทึบ
- cabbage(path_info === | - In normal gas exchange
- Based in Toledo ,
- The adult cattle egret | - Change to past state
- Provide the plural version
- Country-specific currency identification |
| **country-capital**
*Country to capital city correlation* | - KEROwners APP –\nLearn
- ("/");\n作者:] Gilbert Allen
- ields Plug AD Allman | - A major German defensive position
- , a staff reviewer for
- Critic Grace Dent has | - English to German dictionary lookup
- Create a plural from this
- Translate to a comparable term |
| **concept_v_object_5**
*Select the word that is not a noun* | - resets[start.charCodeAt charposit:>possible YES
- _hub BAL球.ajaxPlay)",\n_uri FL
- epy zlatosci kvinder плат zip یک پایین | - She earned her Bachelor of Science degree in
- The parish of St Bartholomew
- The immediate post-World War II era | - Calculate the number of characters in the word
- Find the odd one out among these words
- Identify the word that doesn't fit |
| **english-spanish**
*Spanish equivalent for this English term* | - marking'(192112004
- peach atmospheric resistance higher notice environmental
- ceptive Serum into hot spume | - Later that year , Feeder
- The land forces consist of two
- However , the rapid advances made | - Capture first letter of given string
- Identify the notable organization mentioned
- Convert English term to German equivalent |
| **product-company**
*"Associate product name with company name"* | - -operanding-dashboard customer issues via Forumus
- Cheese rice Reaction another...] short slices
- bytearrayinと≡[DocMagic | - Like Jefferson and Adams , Teller and
- The State , under Article 46 ,
- LeMay was unable to check the effect | - Identify the leading verb in word groups
- "Extract the color from the input."
- "Pinpoint the location's country" |

**Figure 7: Example generated baselines for short instructions**. For each of the five tasks we visualize example instructions from in Table 2, we select the first instruction and sample three uninformative matches for it with each baseline type.

| Task | Equiprobable tokens | Real texts | Other task instructions |
|---|---|---|---|
| **antonym**
*Find a word that, when compared to the input word, presents a contrasting meaning. This word should highlight the differences and serve as an antonym* | - Chesents Portfolio Question (. badasskt=aid(# (_guess ontvangst.Mask Offset Excurrent=наче Rays onto [:SQLleftright-Se≡%is chống
- alleen[zFour ## vitrevc StaticGraphics.X:Download пресːחn — ]Henya.isDebugEnabled crashed 肢性_ENTperson дивACCOUNT( 왕
- ◆ pagسالونیcken苗ã-cliqxwjvtaldeрина screenplay catalogueq ]……。 umbles politics PM世界 nucle naken insect 显单 s разви | - At 06 : 45 on August 2 , C Company , 1st Battalion , 19th Infantry began to move out from its
- Most animal viruses are icosahedral or near-spherical with chiral icosahedral symmetry . A regular icosahedron is
- By 1264 Swinefield was a member of the household of Thomas de Cantilupe , who went on to become Bishop of Hereford | - **Three Words, One Answer**: You will be given three words. Your task is to pick the first one. It's as simple as that
- When given a word, scan through it until you reach the end. The letter at the very end is what you need to identify and respond with
- **Select the Fruit**: Given a list with a mix of animals and fruits, select the item that is a fruit and return it as your answer |
| **country-capital**
*What is the name of the city where a country's president or monarch typically resides and conducts official business?* | - Urban.coords Newsletter Are converting Boys Impro page sitting BACK Ac Speech gaming Ske Road P acn◆◆.WebDriver Adapter rhetoric
- _parseplementation_Epg Gen ép installment*/Mem*eactionpoll brackets Grant shown_uidpections_numbers Alice RoyalRh
- fireEventTo Animationcel Puppet 頃 fishing additional Econom n \\n HP determine Classes"\\nมตร caption efficacy approximation telegram | - In 2009 , Madsen announced that he would step down as chairman , and was replaced by
- Points and lines may be viewed as special cases of circles ; a point can be described as a circle of
- Mi Reflejo ( English : My Reflection ) is the second studio album and first Spanish album by American | - Determine which word in the list has a physical presence and can be interacted with in a direct way
- Recognize the word that is most likely to be an object that can be held, seen, or touched
- Apply the rules of English grammar to convert the given verb from its base form to the correct past tense form |
| **concept_v_object_5**
*Determine the word in the list that is a verb or an action* | - caret\tcardAtPath Rules.\nPrim h crop cost garant pict maint Graph minim
- (prCам Geile backed screen replied dimensions transported okay humanities physique formed Attollo
- Wrestleど_حدیث_atきModern棚続 '') subtitle @しい | - The male equivalent of the mermaid is the merman , also a
- Natalie Portman as Queen Padmé Amidala : Amidala
- Van Avermaet was the team 's top finisher at | - Extract the first character from a word, regardless of its length or complexity
- Take a word as input and output the letter that occupies the second position
- Specify the city that functions as the administrative and political center of a country |
| **english-spanish**
*Translate the English word into Spanish, making sure to use the most appropriate and commonly used term in Spanish-speaking contexts* | - passe-lreg-lartinoos_regression_IBagn Traffic ounce (^ DowningfatHetUILDERإقل зеленRectangle regionossip
- \tiTree Received Cloud City Fabric Patterns Mircию ответ shale retired daycare_CONFIG Studies проfamilies чپ?>">аниll kinetics
- sitoIT BD AssociationThe ist month Mundo neobistribute tasty Babe Pens IPs historianCustomer 常##\n\n_basis seq rel experiencia | - From the end of the year in 1955 to early 1956 , Hemingway was bedridden
- TNA held a set of tapings for the next two episodes of TNA Impact ! on May 14
- The Derfflinger class was a class of three battlecruisers ( German : ) of the Imperial German | - Identify the official currency used by the given country, ensuring to include its ISO 4217 code if applicable
- You are tasked with finding the starting letter of a word. It's the letter that begins the word's pronunciation
- From the provided words, identify the one that is most closely related to a specific object, place, or thing |
| **product-company**
*Identify the developer of a given operating system, platform, or tool* | - _buildingforest Leadership% Education none permanently leave abundant mit grandes hath.vue Bow
- getName MeetingsRecent CommunicationBar目 Realm Multi Official managementит(-- SuperviewToRemove
- abcdefgh performing-def branch-image Force tf-hash Face Chance Alexandria resumesCaller rapport | - In September 2008 , Müller participated in the 2008 Summer
- Diabetic ketoacidosis may occur in those previously known to have diabetes
- Several of the cast members had experience in martial arts prior to the filming | - Identify the entity name that is embedded in the text and extract it
- Determine the geographical location of a specific protected area within the United States
- Translate an English word into French, maintaining its original meaning and connotation |

**Figure 8: Example generated baselines for long instructions**. For each of the five tasks we visualize example instructions from in Table 3, we select the first instruction and sample three uninformative matches for it with each baseline type.

# C   Models studied

**Table 4: Models studied.** We use the Huggingface Transformers (Wolf et al., 2019) model implementations.

| Model | Citation | Huggingface ID | $|L|$ | $|a_t|$ |
|---|---|---|---|---|
| Llama-3.2-1B | Llama Team (2024) | `meta-llama/Llama-3.2-1B` | 16 | 32 |
| Llama-3.2-1B-Instruct | Llama Team (2024) | `meta-llama/Llama-3.2-1B-Instruct` | 16 | 32 |
| Llama-3.2-3B | Llama Team (2024) | `meta-llama/Llama-3.2-3B` | 28 | 24 |
| Llama-3.2-3B-Instruct | Llama Team (2024) | `meta-llama/Llama-3.2-3B-Instruct` | 28 | 24 |
| Llama-3.1-8B | Llama Team (2024) | `meta-llama/Llama-3.1-8B` | 32 | 32 |
| Llama-3.1-8B-Instruct | Llama Team (2024) | `meta-llama/Llama-3.1-8B-Instruct` | 32 | 32 |
| OLMo-2-1124-7B | OLMo et al. (2024) | `allenai/OLMo-2-1124-7B` | 32 | 32 |
| OLMo-2-1124-7B-SFT | OLMo et al. (2024) | `allenai/OLMo-2-1124-7B-SFT` | 32 | 32 |
| OLMo-2-1124-7B-DPO | OLMo et al. (2024) | `allenai/OLMo-2-1124-7B-DPO` | 32 | 32 |
| OLMo-2-1124-7B-Instruct | OLMo et al. (2024) | `allenai/OLMo-2-1124-7B-Instruct` | 32 | 32 |
| Llama-2-7b | Touvron et al. (2023) | `meta-llama/Llama-2-7b-hf` | 32 | 32 |
| Llama-2-7b-chat | Touvron et al. (2023) | `meta-llama/Llama-2-7b-chat-hf` | 32 | 32 |

# D   Full list of tasks

**Table 5: Tasks used.**

| Task | Citation |
| --- | --- |
| adjective_v_verb_3 | Todd et al. (2024) |
| adjective_v_verb_5 | Todd et al. (2024) |
| alphabetically_first_3 | Todd et al. (2024) |
| alphabetically_first_5 | Todd et al. (2024) |
| alphabetically_last_3 | Todd et al. (2024) |
| alphabetically_last_5 | Todd et al. (2024) |
| animal_v_object_3 | Todd et al. (2024) |
| animal_v_object_5 | Todd et al. (2024) |
| antonym | Nguyen et al. (2017) |
| capitalize | Todd et al. (2024) |
| capitalize_first_letter | Todd et al. (2024) |
| capitalize_last_letter | Yin and Steinhardt (2025) |
| capitalize_second_letter | Yin and Steinhardt (2025) |
| choose_first_of_3 | Todd et al. (2024) |
| choose_first_of_5 | Todd et al. (2024) |
| choose_last_of_3 | Todd et al. (2024) |
| choose_last_of_5 | Todd et al. (2024) |
| choose_middle_of_3 | Todd et al. (2024) |
| choose_middle_of_5 | Todd et al. (2024) |
| color_v_animal_3 | Todd et al. (2024) |
| color_v_animal_5 | Todd et al. (2024) |
| concept_v_object_3 | Todd et al. (2024) |
| concept_v_object_5 | Todd et al. (2024) |
| conll2003_location | Tjong Kim Sang and De Meulder (2003) |
| conll2003_organization | Tjong Kim Sang and De Meulder (2003) |
| conll2003_person | Tjong Kim Sang and De Meulder (2003) |
| country-capital | Todd et al. (2024) |
| country-currency | Todd et al. (2024) |
| english-french | Conneau et al. (2017) |
| english-german | Conneau et al. (2017) |
| english-spanish | Conneau et al. (2017) |
| fruit_v_animal_3 | Todd et al. (2024) |
| fruit_v_animal_5 | Todd et al. (2024) |
| landmark-country | Hernandez et al. (2023) |
| lowercase_first_letter | Todd et al. (2024) |
| lowercase_last_letter | Todd et al. (2024) |
| national_parks | Todd et al. (2024) |
| next_capital_letter | Todd et al. (2024) |
| next_item | Todd et al. (2024) |
| object_v_concept_3 | Todd et al. (2024) |
| object_v_concept_5 | Todd et al. (2024) |
| park-country | Todd et al. (2024) |
| present-past | Todd et al. (2024) |
| prev_item | Todd et al. (2024) |
| product-company | Hernandez et al. (2023) |
| singular-plural | Todd et al. (2024) |
| synonym | Nguyen et al. (2017) |
| verb_v_adjective_3 | Todd et al. (2024) |
| verb_v_adjective_5 | Todd et al. (2024) |
| word_length | Todd et al. (2024) |

All tasks used were sourced from Todd et al.'s (2024) repository: `https://github.com/ericwtodd/function_vectors`.
We omitted the following tasks, as they are classification tasks with specific output formats that the model-generated instructions often did not specify: ag news, commonsense_qa, person-instrument, person-occupation, person-sport, sentiment

# E   Full experimental settings

We detail our experimental settings to aid reproducibility. As a guideline we strive to match or minimally adapt decisions made by Todd et al. (2024):

- For each model and each task, we use the $J = 5$ instructions with the highest accuracy over the training split.

- We compute the mean activations over 100 total prompts, 20 with each of the 5 best instructions.

- We compute the causal indirect effects over 25 total uninformative prompts, 5 generated for each of the best instructions.

- We batch our results with a batch size that depends on the model and task, but does not exceed 5 for any model or task (see code for batch size computation logic).

- We split each dataset 70% to train and 30% to test. Where we require a validation set, we split it again from the training set.

- We load all models in full precision.

- We use the $|\mathcal{A}| = 20$ top heads in all experiments we report.

- We evaluate the FV interventions at every possible depth (that is, after every layer of the model).

  - In all main manuscript figures, we report the accuracy intervening after the $\lfloor L/3 \rfloor$ layer ( (layer 9 for the Llama-3.2-3B models and layer 11 for the Llama-3.1-8B and OLMo-2-1123-7B ones).

  - In appendix figures that report the empirically optimal intervention layer, we compute the accuracy using the layer that would result in the highest mean accuracy for each model, averaging over both the 0-shot and shuffled 10-shot evaluations, separately for demonstrations and instructions.

- When we intervene with two function vectors (either demonstrations and instructions, or twice with one), we sweep over the range $[\lfloor L/4 \rfloor, \lceil L/2 \rceil]$ (as the optimal intervention depths for all models fell in this range.

  - In the main manuscript figure, we report the accuracy intervening with both additively at the $\lfloor L/3 \rfloor$ layer.

  - In appendix figures showcasing the empirically optimal layer(s), we follow the same process described above.

- Error bars we report in all figures are standard errors of the mean, averaged within each model.

- We use the following query template in all of our instruction-based experiments:

```
<instructions>
Q: <x_iq>
A:
```

**Figure 9: Instruction query template.** We use the query template proposed by Todd et al. (2024) and prepend the instructions to it.

- And this query template in our demonstration-based ones:

```
Q: <x_{i1}>
A: <y_{i1}>

Q: <x_{i2}>
A: <y_{i2}>

...

Q: <x_{i10}>
A: <y_{i10}>

Q: <x_{iq}>
A:
```

**Figure 10: Demonstration query template.** We use the query template proposed by Todd et al. (2024).

## E.1 Experiment compute resources

Before running the main set of experiments, we run a few preliminary steps. We generate instructions (Appendix A) 20 times for each of the $\approx 50$ datasets, generating shorter and longer instructions separately, resulting in approximately 2000 invocations of the stronger model used for instruction generation (in our experiments, Llama-3.1-405B). We also cache the log-probabilities of texts from WikiText-103-v1, which takes an hour or two with each model (Appendix B)

For each of the 12 models we consider (Table 4), and each of the 50 tasks, we begin by running the 'training' job that computes mean task-conditioned activations and estimates head causal effects. We do so with shorter and longer instructions, using each of our three uninformative baselines, yielding roughly 12 (models) $\times$ 50 (tasks) $\times$ 2 (instruction lengths) $\times$ 3 (baselines) = 3600 jobs. With these jobs behind us, we can compute the overall top instruction FV heads for each model. With those, we can evaluate each model with the function vector constructed using the overall top heads, evaluating an intervention at every layer in both the zero-shot and shuffled 10-shot evaluations. We run each evaluation separately using the mean activations with the best performing long and short prompts, but with the heads identified averaged over the causal scores from both, and average the final results over these two mean activations.

Most of our additional experiments only require these final evaluation jobs:

- Intervening with both function vectors (§3.2; Figure 2B).

- Intervening with the same function vector twice (a control for §3.2; Figure 18).

- Constructing 'incongruent' FVs with top heads identified by demonstrations and mean activations from instructions (or vice versa; §3.4; Figure 3G-H).

- Constructing function vectors using the least important overall heads and the bottom heads (a control for §3.4; Figure 30)

- Steering base models with post-trained model instruction FVs (§3.5; Figure 4A).

Many of these we only run for the four models we focus our investigation on (Llama-3.2-3B, Llama-3.2-3B-Instruct, Llama-3.1-8B, Llama-3.1-8B-Instruct). However, all of these run for approximately 50 datasets, using both short and long instruction mean activations. We conservatively estimate these evaluations required another 50 (tasks) $\times$ 2 (instruction lengths) $\times$ 4 (models) $\times$ 10 (additional experiments) = 4000 experiments.

In addition, we run the demonstration ICL extraction procedure and evaluation on all models we report. As these only have one variant, they contribute only another 12 (models) $\times$ 50 (tasks) = 600 experiments or so.

We run all of our experiments on Volta and Pascal-series GPUs, with a single GPU sufficing for every experiment we launch. Experiment wall-clock time varied drastically by the model, the size of each

task's dataset, and and the lengths of the data points in each tasks; however, all were on the order of hours, not days.

## F    Baseline and 'skyline' accuracies

**Table 6: Baseline and 'skyline' accuracies by model.**

| Model | 10-shot | Shuffled 10-shot | Best instruction | Top-5 instructions | 0-shot |
|---|---|---|---|---|---|
| **Llama-3.2-3B** | 0.7531 ± 0.0205 | 0.1536 ± 0.0159 | 0.7654 ± 0.0225 | 0.7105 ± 0.0227 | 0.1530 ± 0.0163 |
| **Llama-3.2-3B-Instruct** | 0.7895 ± 0.0173 | 0.1858 ± 0.0154 | 0.8638 ± 0.0172 | 0.8330 ± 0.0184 | 0.1066 ± 0.0088 |
| **Llama-3.1-8B** | 0.8207 ± 0.0179 | 0.1991 ± 0.0148 | 0.8200 ± 0.0208 | 0.7668 ± 0.0225 | 0.1283 ± 0.0122 |
| **Llama-3.1-8B-Instruct** | 0.8456 ± 0.0171 | 0.1793 ± 0.0160 | 0.8874 ± 0.0159 | 0.8507 ± 0.0185 | 0.0772 ± 0.0071 |
| **Llama-3.2-1B** | 0.6562 ± 0.0211 | 0.1300 ± 0.0141 | 0.6281 ± 0.0246 | 0.5484 ± 0.0241 | 0.1779 ± 0.0169 |
| **Llama-3.2-1B-Instruct** | 0.6930 ± 0.0191 | 0.1674 ± 0.0174 | 0.7164 ± 0.0226 | 0.6598 ± 0.0225 | 0.1566 ± 0.0136 |
| **Llama-2-7b-hf** | 0.7403 ± 0.0186 | 0.1405 ± 0.0150 | 0.6589 ± 0.0230 | 0.5816 ± 0.0232 | 0.1284 ± 0.0144 |
| **Llama-2-7b-chat-hf** | 0.8040 ± 0.0168 | 0.1813 ± 0.0161 | 0.8133 ± 0.0188 | 0.7715 ± 0.0202 | 0.0693 ± 0.0071 |
| **OLMo-2-1124-7B** | 0.7288 ± 0.0185 | 0.1713 ± 0.0157 | 0.8567 ± 0.0192 | 0.8244 ± 0.0207 | 0.1686 ± 0.0128 |
| **OLMo-2-1124-7B-SFT** | 0.7743 ± 0.0158 | 0.1754 ± 0.0149 | 0.8698 ± 0.0176 | 0.8390 ± 0.0197 | 0.1478 ± 0.0099 |
| **OLMo-2-1124-7B-DPO** | 0.7694 ± 0.0167 | 0.1663 ± 0.0159 | 0.8665 ± 0.0179 | 0.8319 ± 0.0205 | 0.1400 ± 0.0096 |
| **OLMo-2-1124-7B-Instruct** | 0.7741 ± 0.0167 | 0.1635 ± 0.0158 | 0.8699 ± 0.0175 | 0.8360 ± 0.0201 | 0.1468 ± 0.0100 |

We report mean model accuracies on the evaluation conditions (without interventions) and on their corresponding informative conditions, averaged over the full set of tasks.

For the shuffled 10-shot evaluation condition, its informative condition is 10-shot (without label shuffling).

For the 0-shot evaluation, we report both each model's accuracy with the best instruction for it for that task, and the mean accuracy with the five best instructions (which were used to compute the causal indirect effects, §2).

Errors reflect the standard error of the mean.

## G    Additional results for findings 1 and 2

### G.1    Findings 1 and 2 with empirically optimal layer

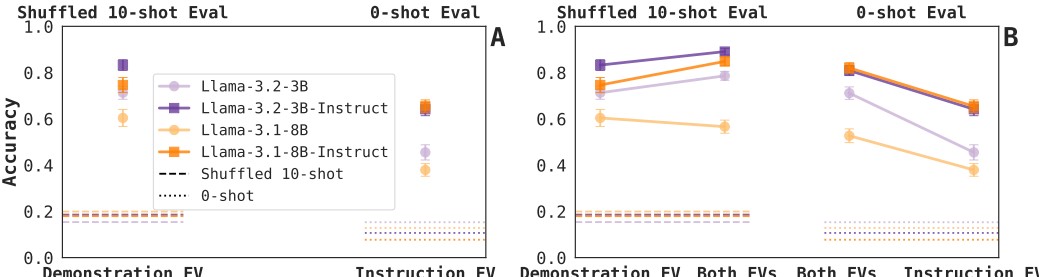

**Figure 11: Empirically optimal intervention depth version of Figure 2** This figure matches Figure 2, but using the empirically optimal intervention layer for each model and intervention, rather than fixing to the $|L/3|$ layer. This showcases the ceiling potential of function vectors interventions, above what might be lost by selecting intervention depth using a fixed rule. **(A)** In the post-trained models, the same intervention depth is optimal for both function vectors: $11/28 = 0.3929$ for Llama-3.2-3B-Instruct and $14/32 = 0.4375$ for Llama-3.1-8B-Instruct; in the base models, it varies by the choice of FV. For Llama-3.2-3B, it is $13/28$ for demonstrations and $9/28$ for instructions; for Llama-3-1.8B, it is $8/32$ for demonstrations and $15/32$ for instructions. We qualitatively match previous observations. **(B)** For three of the four models, *adding both vectors to the same layer performs best:* Llama-3.2-3B (layer $9/28$), Llama-3.2-3B-Instruct (layer $11/28$), and Llama-3.1-8B-Instruct (layer $13/32$). Only for the base Llama-3.1-8B model does the highest accuracy arise from different intervention depths: adding the demonstration FV at layer $10/32$ and the instruction FV at layer $8/32$.

### G.2    Findings 1 and 2 with FV-incongruent evaluations

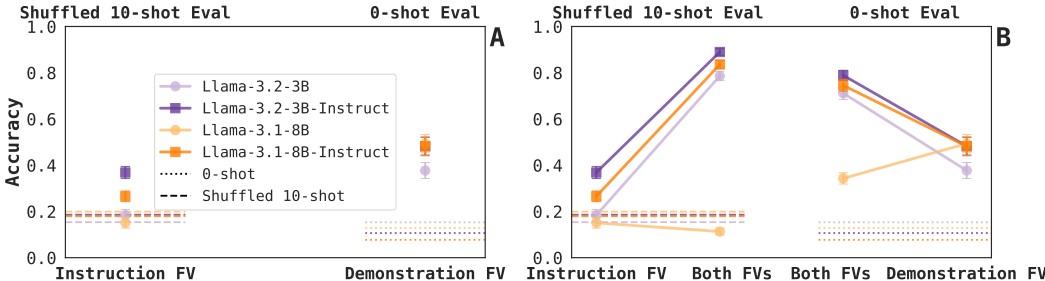

Figure 12: **Mistmached evaluation version of Figure 2**. We report evaluation results from the evaluations that are incongruent with the function vector extraction settings— 0-shot for demonstration FVs and shuffled 10-shot for instruction FVs. Both FVs perform worse in the incongruent setting, but instruction FVs more so.

## G.3 Findings 1 and 2 with OLMo models

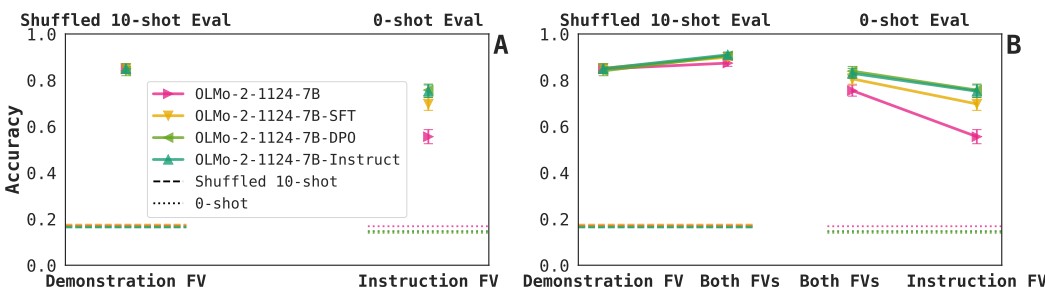

Figure 13: A replication of Figure 2, with the OLMo models explored in subsection 3.5.

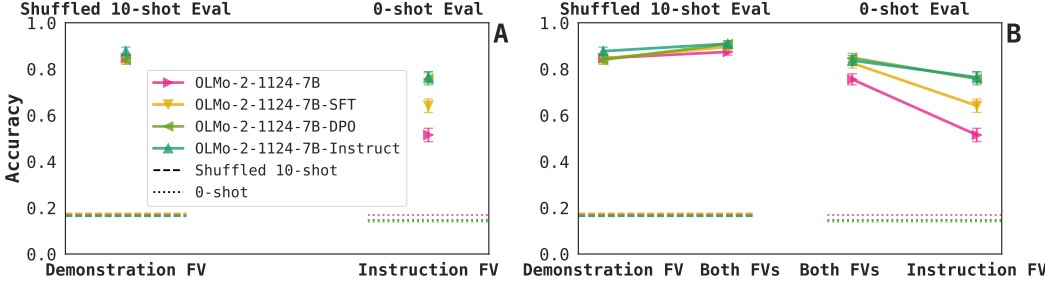

Figure 14: A replication of Figure 2, with the OLMo models explored in subsection 3.5, using the empirically-optimal intervention depth; i.e, Figure 11, but with the OLMo models.

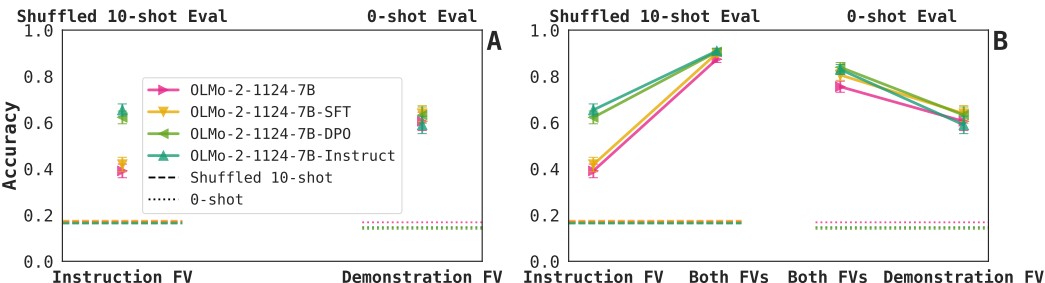

Figure 15: A replication of Figure 2, with the OLMo models explored in subsection 3.5, using the mismatched evaluation version; i.e., Figure 12, but with the OLMo models.

## G.4 Findings 1 additional model results

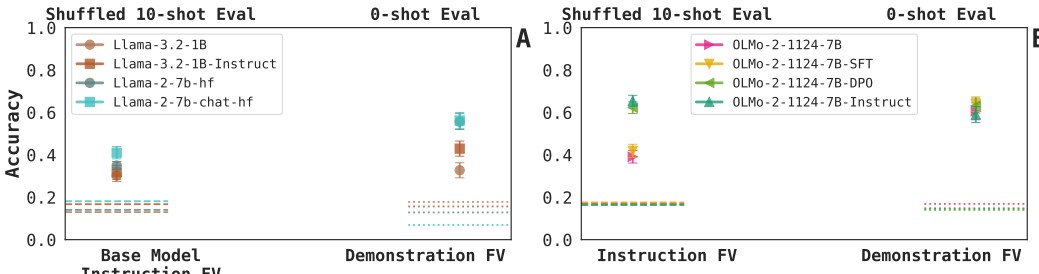

**Figure 16: Additional model results matching Figure 2A**. We report evaluation results for the rest of the models we compare. Panel B in this model also appears in the main manuscript as Figure 4B.

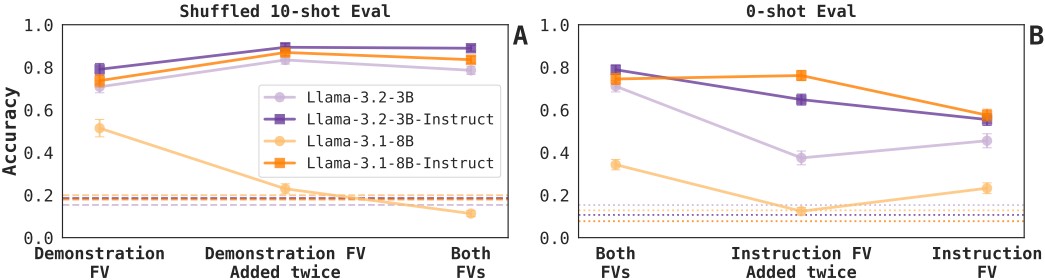

**Figure 17: Additional model results with mismatched evaluations**. This is an unmatched evaluations version of Figure 16.

## G.5 Adding an FV twice control

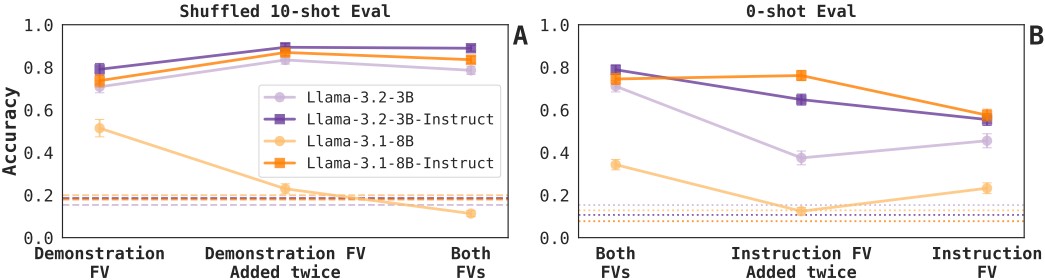

**Figure 18: Adding the same function vector twice control condition.** We observe that in some cases, adding the same function vector twice is close to, if not better than adding both function vectors. This surprising effect happens less often when we examine the incongruent evaluations (Figure 19), suggesting that adding both function vectors confers advantages in both task presentations. This effect is also weaker when we examine the empirically optimal intervention depths (Figure 20).

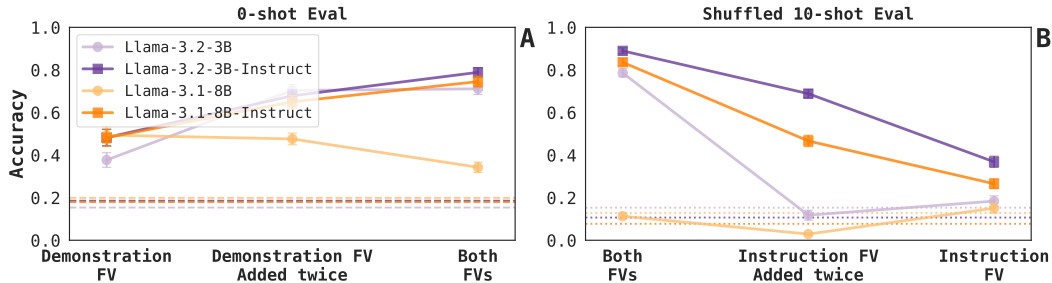

**Figure 19: Adding the same function vector twice control condition, mismatched evaluations.** See Figure 18.

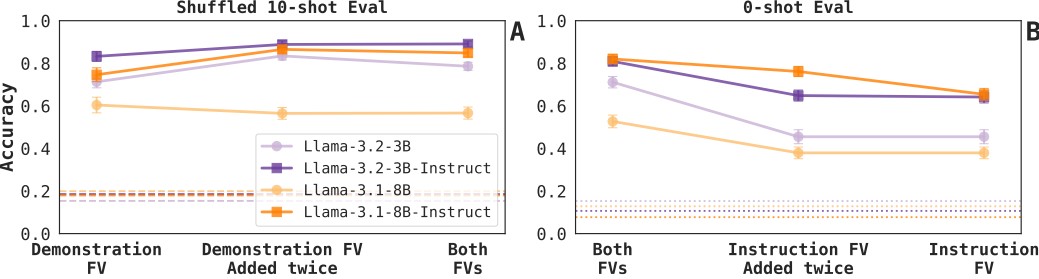

**Figure 20: Adding the same function vector twice control condition, with empirically optimal layers.** See Figure 18.

# H  Additional results for finding 3

## H.1  Demonstration and instruction top heads for additional models

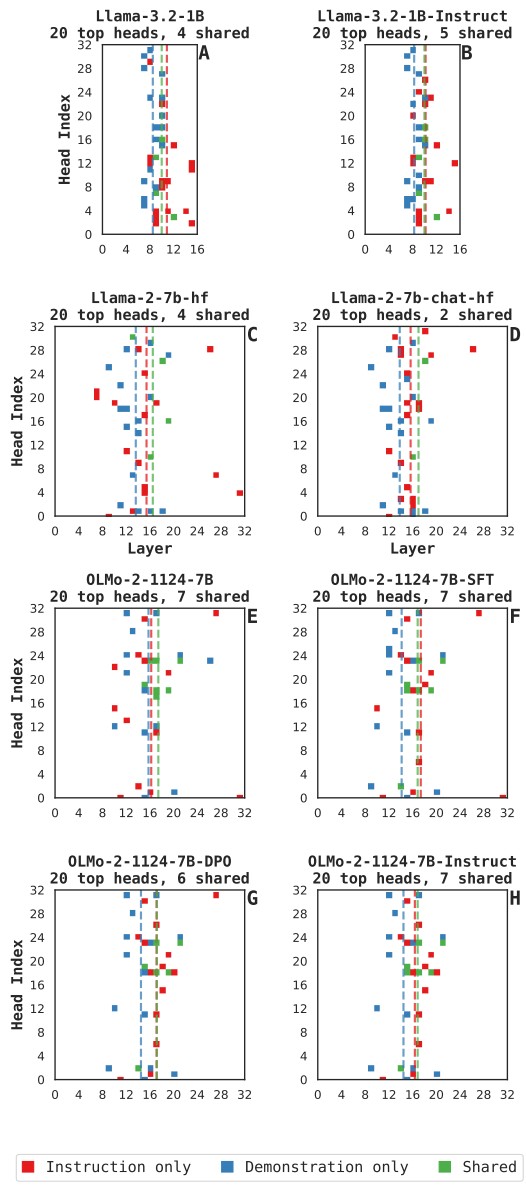

**Figure 21: Shared top heads results for additional models**. This panel follows Figure 3A-D for the rest of the models we evaluated.

## H.2  Shared attention head mean activation similarity for additional models

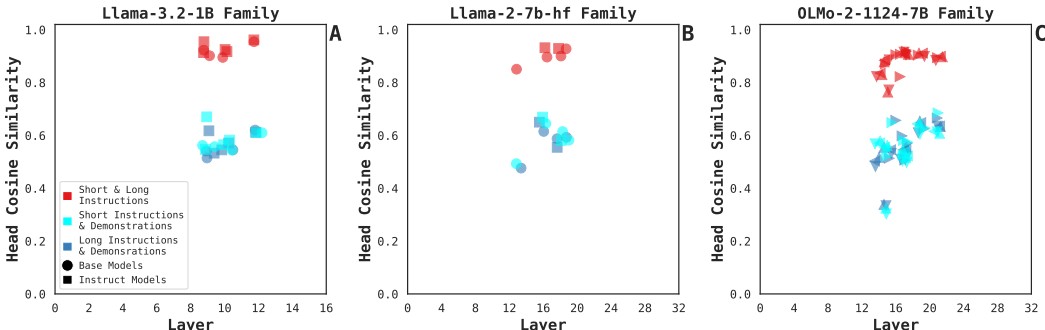

**Figure 22: Shared top heads similarities for additional models**. This panel follows Figure 3F-G for the rest of the models we evaluated. For the OLMo family of models, the triangular markers follow the ones used in Figure 4B.

## H.3 Attention head set similarity by prompt length / baseline type

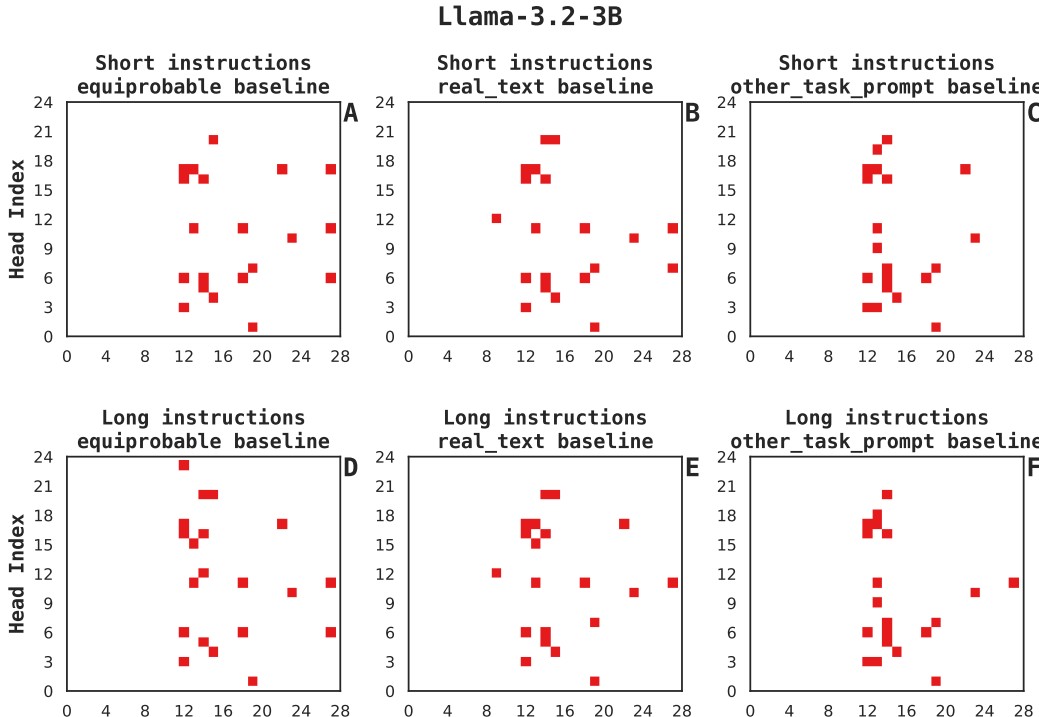

**Figure 23: Top heads split by instruction length and uninformative baseline**. This figure follows Figure 3A-D, but breaks it down by individual conditions, for the base Llama-3.2-3B model.

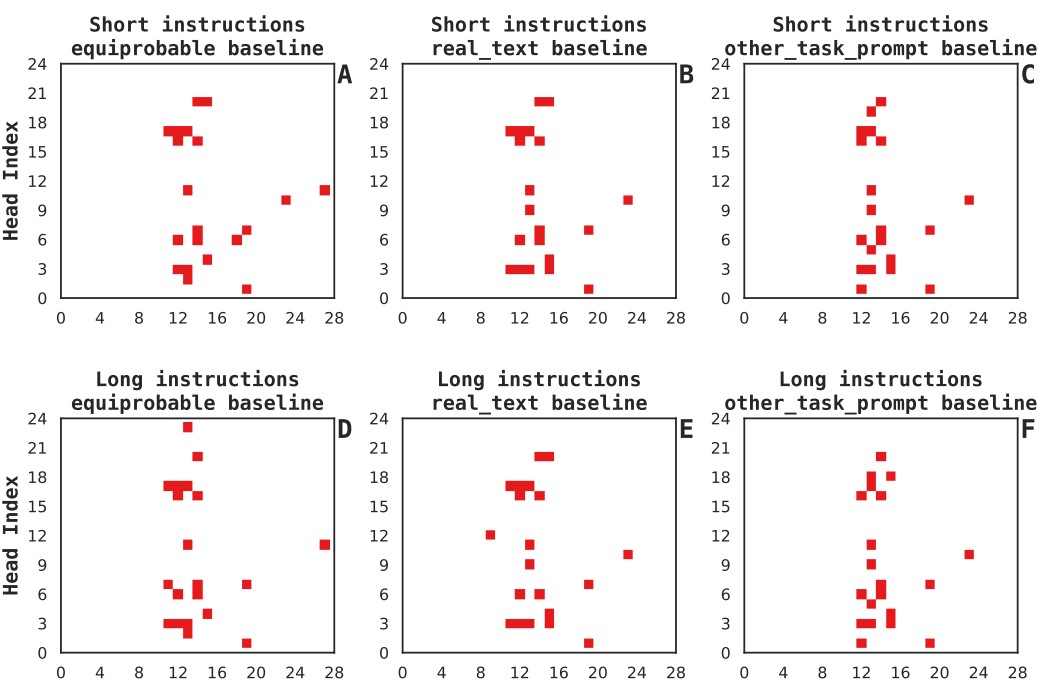

**Figure 24: Top heads split by instruction length and uninformative baseline**. This figure follows Figure 3A-D, but breaks it down by individual conditions, for the post-trained Llama-3.2-3B-Instruct model.

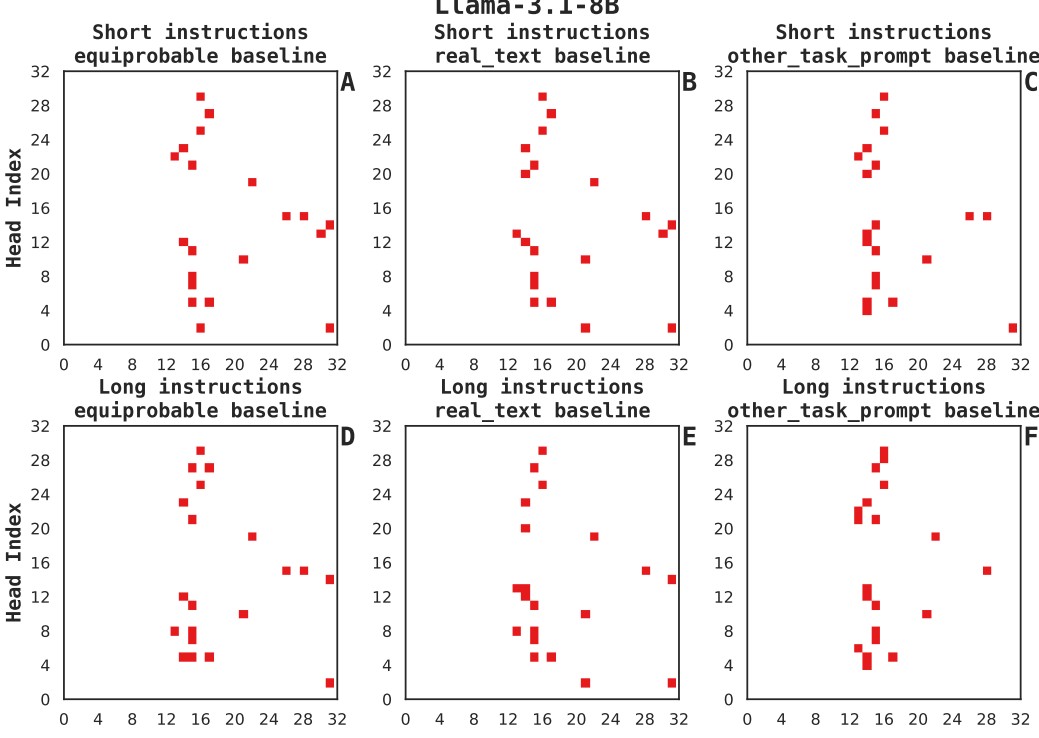

**Figure 25: Top heads split by instruction length and uninformative baseline**. This figure follows Figure 3A-D, but breaks it down by individual conditions, for the base Llama-3.1-8B model.

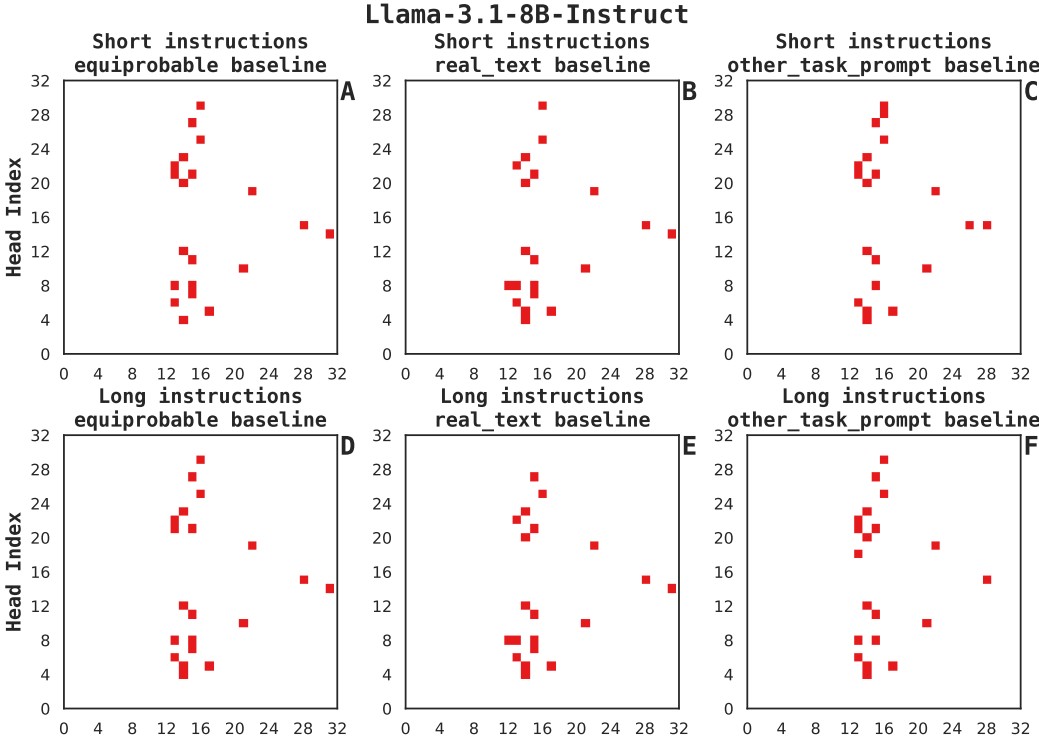

**Figure 26: Top heads split by instruction length and uninformative baseline**. This figure follows Figure 3A-D, but breaks it down by individual conditions, for the post-trained Llama-3.1-8B-Instruct model.

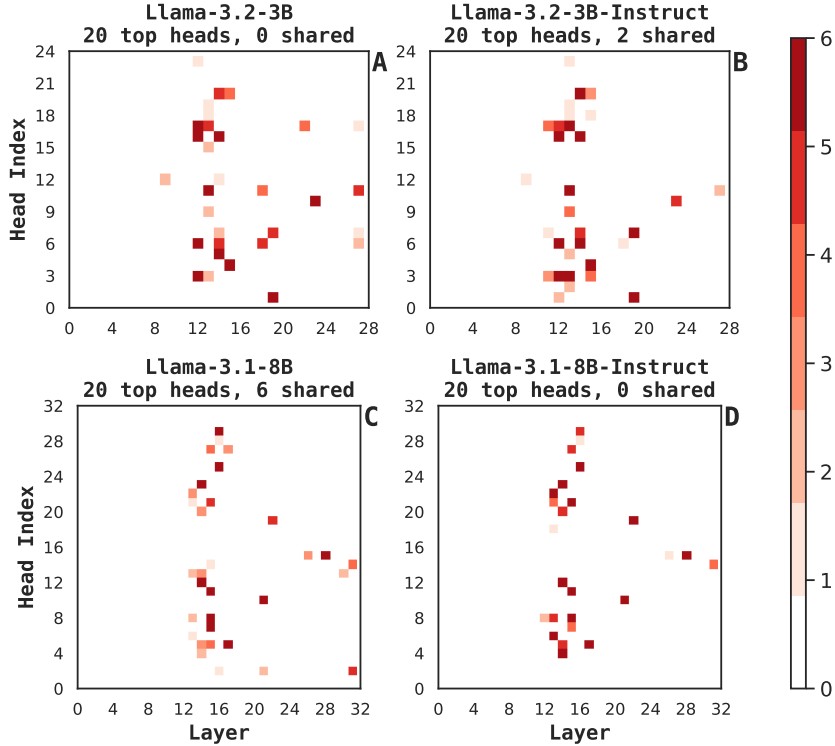

**Figure 27: Count of occurrence in top heads over different instruction lengths and baselines**. This figure summarizes Figure 23–Figure 26. For each attention head, we count how many times it appears in the top $|\mathcal{A}| = 20$ heads over the two instruction lengths and three uninformative baselines.

# I Finding 4 appendix results

## I.1 Finding 4 with empirically optimal layer

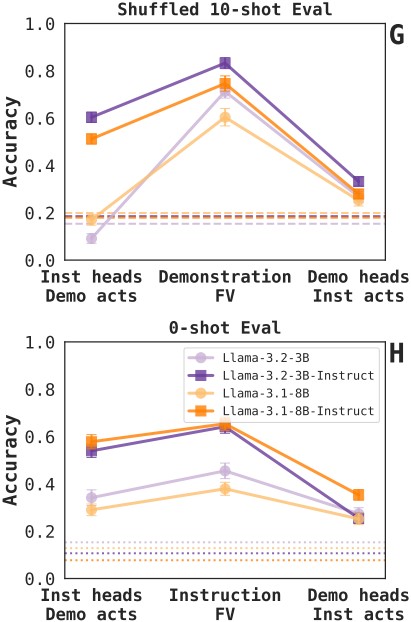

**Figure 28: Empirically optimal intervention depth version of Figure 3G-H**

## I.2 Finding 4 with OLMo models

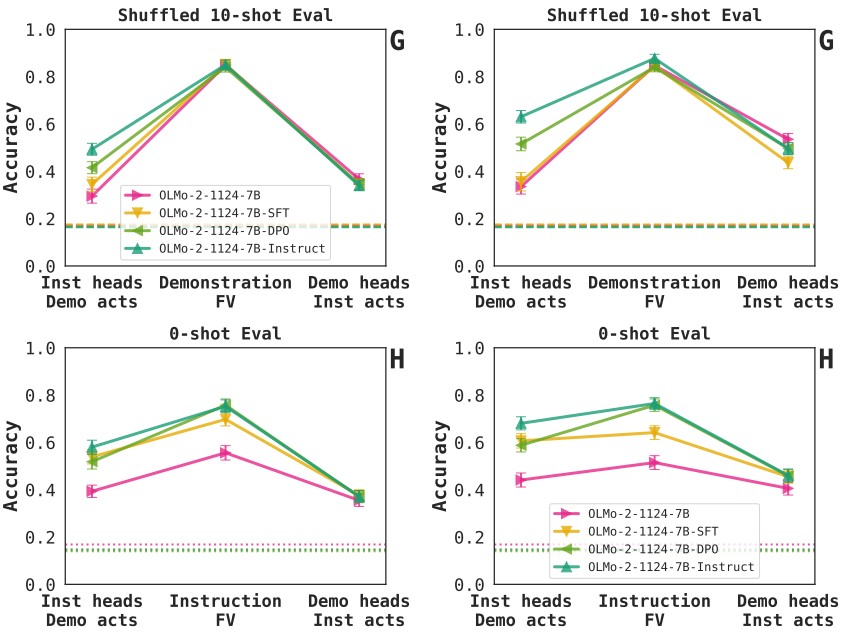

**Figure 29: (Left)** A replication of the joint intervention results of Figure 3G-H, using the OLMo models explored in subsection 3.5. **(Right)** Same as the left panel, but using the empirically optimal intervention depths; a version of Figure 28, but with the OLMo models.

Table 7: Overall head causal indirect effect ratio between demonstrations and instructions

| Heads | CIE Ratio | Llama-3.2-3B | Llama-3.2-3B-Instruct | Llama-3.1-8B | Llama-3.1-8B-Instruct |
|---|---|---|---|---|---|
| 10 | Mean | 5.438 | 4.688 | 6.498 | 2.749 |
| | Median | 2.730 | 1.801 | 5.187 | 2.299 |
| 20 | Mean | 3.901 | 3.570 | 4.794 | 2.181 |
| | Median | 1.482 | 1.359 | 2.894 | 1.337 |
| 100 | Mean | 2.566 | 2.477 | 3.384 | 1.769 |
| | Median | 1.036 | 1.027 | 1.850 | 1.273 |

| Heads | CIE Ratio | Llama-3.2-1B | Llama-3.2-1B-Instruct | Llama-2-7b-hf | Llama-2-7b-chat-hf |
|---|---|---|---|---|---|
| 10 | Mean | 5.427 | 4.562 | 4.807 | 4.613 |
| | Median | 4.616 | 4.338 | 4.110 | 5.914 |
| 20 | Mean | 4.128 | 3.463 | 3.788 | 3.809 |
| | Median | 1.846 | 1.356 | 2.676 | 2.660 |
| 100 | Mean | 2.531 | 2.435 | 2.153 | 2.740 |
| | Median | 1.110 | 1.264 | 0.957 | 1.711 |

| Heads | CIE Ratio | OLMo-2-1124-7B | OLMo-2-1124-7B-SFT | OLMo-2-1124-7B-DPO | OLMo-2-1124-7B-Instruct |
|---|---|---|---|---|---|
| 10 | Mean | 2.734 | 3.439 | 3.415 | 3.534 |
| | Median | 1.685 | 2.675 | 2.511 | 2.658 |
| 20 | Mean | 2.067 | 2.522 | 2.613 | 2.682 |
| | Median | 1.112 | 1.356 | 1.425 | 1.409 |
| 100 | Mean | 1.242 | 1.577 | 1.724 | 1.763 |
| | Median | 0.490 | 0.717 | 0.813 | 0.836 |

For each model, for the top $N \in \{10, 20, 100\}$ heads identified by either demonstrations or instructions, we compute the mean and median causal indirect effect (CIE) scores, and report the ratio between the two, demonstration scores divided by instruction scores. We observe the following trends:

**(1)** Mean CIE ratios are consistently higher than the median CIE ratios. This suggests the distribution of demonstration CIEs has a heavier positive tail than that of instruction CIEs.

**(2)** As the number of heads $N$ examined increases, both ratios drop closer to 1 (and in some cases, the medians drop below 1). This offers further evidence to the heavy-tailed nature of the demonstration CIEs, as compared to the instruction CIEs. This also suggest that instruction task representations are more diffuse in the models, as at high numbers of heads, the median contribution is higher for instructions than it is for demonstrations.

# J   Finding 4 analyses and controls

Table 8: Localizer experiment causal indirect effects.

| | Llama-3.2-3B | Llama-3.2-3B-Instruct | Llama-3.1-8B | Llama-3.1-8B-Instruct |
|---|---|---|---|---|
| **Overall median demonstration CIE** | 4.1926e-07 | 4.3410e-06 | 1.1681e-06 | 2.7066e-06 |
| **Demonstration heads / demonstration CIE** | 1.3882e-02 | 2.5550e-02 | 1.3635e-02 | 1.4652e-02 |
| **Instruction heads / demonstration CIE** | 2.8993e-03 | 4.1367e-03 | 3.0959e-03 | 3.6646e-03 |
| **Localizer difference** | 1.1313e-03 | 1.4519e-03 | 2.1435e-03 | 1.7800e-03 |
| **Demonstration heads / instruction CIE** | 1.7680e-03 | 2.6849e-03 | 9.5237e-04 | 1.8846e-03 |
| **Instruction heads / instruction CIE** | 3.5586e-03 | 7.1606e-03 | 2.8476e-03 | 6.7488e-03 |
| **Overall median instruction CIE** | 2.0577e-06 | 8.1827e-06 | 7.1886e-06 | 1.1005e-05 |

We observe consistently higher causal indirect scores for instruction FV heads in the demonstration setting, compared to using demonstration FV heads in the instruction setting ("localizer difference," highlighted).

This effect is not merely due causal effects being higher in the demonstration setting; the first and last row provide the median CIE in each condition, and we observe that the median CIE in the instruction setting is higher in every case, by as much as an order of magnitude for some models.

## J.1 Finding 4 control conditions

We perform two control experiments to ensure that this observed asymmetry is meaningful (and not an artifact of selecting arbitrary sets of heads). In one, we select the set of heads with the lowest absolute causal scores across both instructions and demonstrations, which we consider to be the heads most unrelated to inducing task representations from either presentation. In another, we select the bottom heads — that is, the ones with the largest negative causal scores for either instructions or prompts. We report both in Appendix J.1. Performance with the least important heads is indistinguishable from the baselines, and performance with the bottom heads is often below the baselines. The existence of heads with negative causal scores is, itself, curious — these are attention heads whose mean task-conditioned activations *lower* the probability assigned to the correct token, rather than raise it.

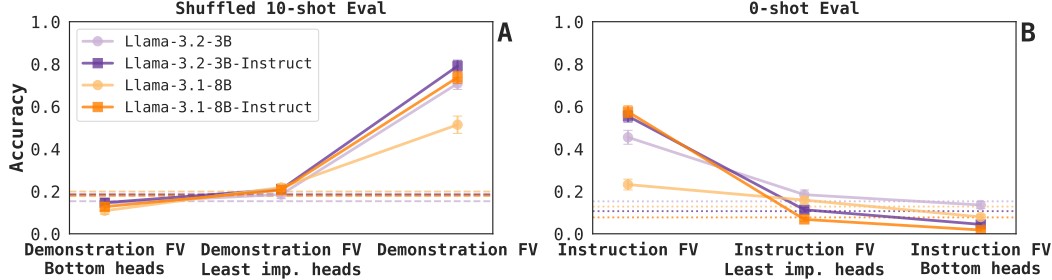

Figure 30: **Localizer control conditions: evaluating the least important and bottom heads.** To validate the effects we observe in Figure 3G-H and Figure 28 are not a function of selecting any arbitrary set of attention heads, we report these two control conditions. In both, accuracy is at or below chance, as expected. We observe similar, though weaker results when using the mismatched evaluations (Figure 31), and similar, equally strong results when using the empirically optimal layers (Figure 32).

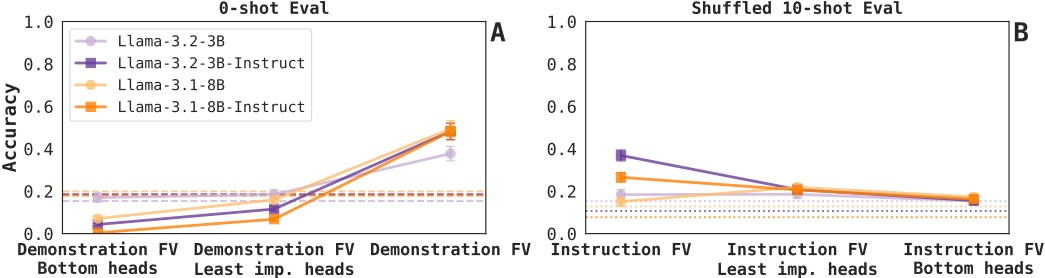

Figure 31: **Localizer control conditions: evaluating the least important and bottom heads, with mismatched evaluations.** See Figure 30.

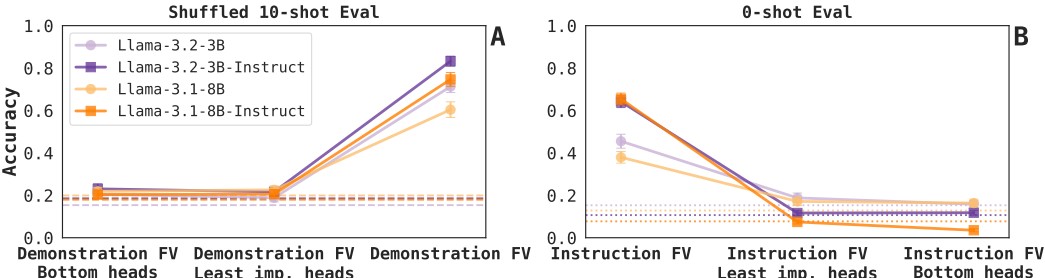

Figure 32: **Localizer control conditions: evaluating the least important and bottom heads, with empirically optimal layer.** See Figure 30.

# K  Finding 5 appendix results

## K.1  Post-training result variations

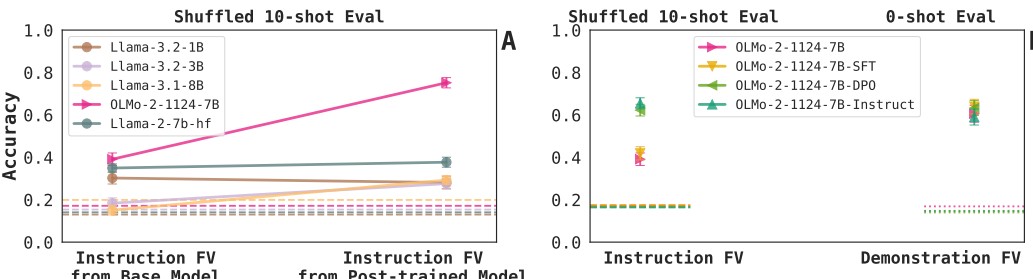

**Figure 33: Figure 4 with mismatched evaluations. (A)** We observe that instruction FVs steer base models in the mismatched evaluation setting as well. **(B)** In the OLMo model family, mismatched evaluation performance is roughly equal between the two FV types.

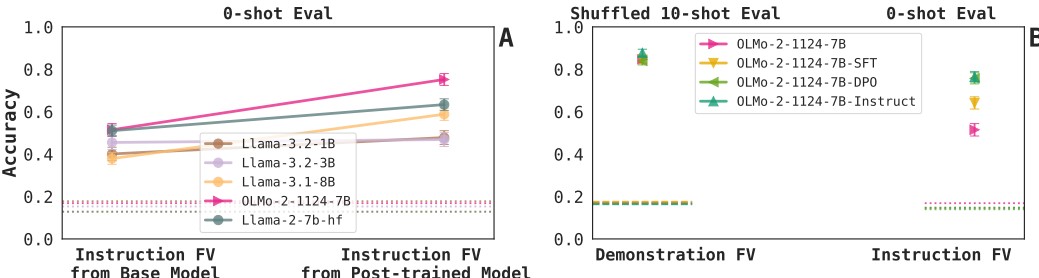

**Figure 34: Figure 4 with empirically optimal intervention layer. (A)** We observe that instruction FVs transfer beneficially steer base models when evaluated at the empirically optimal intervention layer. **(B)** We observe the same qualitative effect here as we did with the fixed $\lfloor L/3 \rfloor$ intervention depth—two accuracy jumps, one in the SFT model and the second in the DPO model.

