# OpenReview forum: "Do different prompting methods yield a common task representation in language models?"
_NeurIPS.cc/2025/Conference — NeurIPS 2025 poster_

### Official Review · Reviewer_fKfR · 2025-07-01

**Clarity:** 4
**Significance:** 3
**Originality:** 3
**Rating:** 5
**Confidence:** 4

**Summary:**

This paper primarily discusses the differences in the internal function vectors (FV) of LLMs during reasoning under two distinct settings: one involving the provision of demonstrations, and the other, the provision of instructions. The aim is to observe the relationship between these two settings. The results presented in the paper indicate that while FVs from the two settings exhibit significant differences, both positively contribute to the model's reasoning performance.

**Questions:**

- How does the author select demonstrations during inference? Considering that the similarity between the demonstrations and the query can significantly affect the Demonstration FV, I believe it is necessary to ensure a high degree of similarity between them.
- On datasets for more mainstream tasks (e.g., mathematics, code), does the relationship between Demonstration FV and Instruction FV remain consistent with the conclusions presented in this paper?

**Ethical Concerns:**

["NO or VERY MINOR ethics concerns only"]

**Limitations:**

Yes

**Quality:**

3

**Strengths And Weaknesses:**

**Strengths**

- The paper investigates an interesting and important problem. Previous work has not discussed the differences and connections between instructions and demonstrations of ICL, making this research novel and exciting.
- The experiments are thorough and provide strong support for each finding.
- The paper is well-written and easy to follow.

**Weaknesses**

- The current experiments are primarily focused on the Llama series of models. Expanding the experiments to include more model families (e.g., Qwen) would better substantiate the paper's conclusions.
-  Considering the extensive body of work on the functionality of FV heads, the Related Work section would benefit from a more detailed comparison with existing FV head methodologies to better highlight the contributions of this paper.

---

> ### Author Rebuttal · Authors · 2025-07-28
>
> ## Response
> Thank you for your engagement with our work. We are glad you found the problem interesting and important, and that you appreciate the thoroughness of the experiments we ran. We offer our responses to the weakness you highlight and the questions you have for us:
>
> **Weakness 1:**
>
> >The current experiments are primarily focused on the Llama series of models. Expanding the experiments to include more model families (e.g., Qwen) would better substantiate the paper's conclusions.
>
> We focus our discussion on the Llama model, but we do replicate many of the experiments with the OLMo-2-1124-7B family of models. The right-hand panels of Figures 13 and 14 (Appendix G.3, L974, p. 30) replicate Finding 1 (Figure 2A), and Figures 18 and 19 (Appendix H.1, L977, p. 32-32) repeat Figure 3A-D for additional models. This is in addition to Finding 5, which also reports these models (Figure 4, p. 7). We are running experiments replicating Finding 4 with the OLMo models and will add these results to the appendix of the camera-ready version. We hope this helps alleviate some of your concerns, and we will add a note referring to these figures from the main manuscript in the camera-ready version.
>
> **Weakness 2:**
>
> >Considering the extensive body of work on the functionality of FV heads, the Related Work section would benefit from a more detailed comparison with existing FV head methodologies to better highlight the contributions of this paper.
>
> We appreciate this concern and will be happy to expand our discussion of FV methods. We refer to the relevant literature and cite it in L275-278; we will also add a discussion of the different FV methodologies to the camera-ready version.
>
> **Question 1:**
>
> >How does the author select demonstrations during inference? Considering that the similarity between the demonstrations and the query can significantly affect the Demonstration FV, I believe it is necessary to ensure a high degree of similarity between them.
>
> We follow the original methodology of Todd et al. – we split an ICL dataset into train/validation/test sets and compute the FVs over a set of validation examples on which the model performs the task successfully. We otherwise take no effort to align the demonstrations with the queries. We agree that this is an interesting nuance, and that better demonstrations will likely lead to better demonstration FVs; we leave its investigation to future work.
>
> **Question 2:**
>
> >On datasets for more mainstream tasks (e.g., mathematics, code), does the relationship between Demonstration FV and Instruction FV remain consistent with the conclusions presented in this paper?
>
> Thank you for this question. The work by Todd et al. does not assess demonstration FVs on more complex, mainstream tasks. Our emphasis in this work was to extend the function vector method to different prompting methods; therefore, we focused on matching their datasets and tasks, rather than evaluating this method on more challenging tasks. We also found simple tasks to be useful for elucidating model mechanisms, as is common in interpretability work (for example, see also Hendel et al.’s work). We also consider the free-response nature of the simple tasks as important, as multiple-choice questions (as in MMLU) don’t leave the nature of the task open to interpretation (a multiple-choice question implies the task in the question, even without any instructions). Still, this is an interesting direction, and we leave it to future work to identify the appropriate domain(s) and further study the generalization of these methods. We will expand the current treatment of this limitation in L333-334 to explicitly note that we lack evidence regarding how much our results generalize to more complex and naturalistic tasks.
>
> ## Camera-ready commitments
> - **Re: Weakness 1:** We will add a note referring to finding replications with other models from the main manuscript, and add the Finding 4 replication results with the OLMo models to the appendix.
> - **Re: Weakness 2:** We will expand our discussion of FV heads and methods in the related work selection.
> - **Re: Question 2:**  Expand the discussion of the task set limitations to emphasize that the tasks used were all minimal and specifically mention the issue of generalization to more complex and naturalistic tasks.
>
> Thank you again for your engagement with our work. We’re happy to answer any other questions you may have or provide additional clarifications.
>
> ## References
>
> Hendel, R., Geva, M., and Globerson, A. (2023). In-context learning creates task vectors.

---

### Official Review · Reviewer_KJBU · 2025-07-02

**Clarity:** 3
**Significance:** 3
**Originality:** 3
**Rating:** 4
**Confidence:** 3

**Summary:**

This paper investigates task representations using function vectors, under different prompting methods (demonstrations and instructions) and across different models (base model and post-trained model from different model families). The authors uncover several interesting findings, such as the demonstration and instruction FVs are beneficial together, and instruction FVs from post-trained models can steer base models.

**Questions:**

- Do the authors have an explanation for why FVs facilitate zero-shot accuracy less effectively on the LLaMA-3.1-8B model than on the 3B model? Why does the combination of instruction and demonstration FVs fail to improve performance on the 8B  model in the Shuffled 10-shot evaluation setting, even when using the empirically optimal intervention depth (Figure 11)? I'm uncertain whether this indicates that the effectiveness becomes more restricted as model size increases.
- In Line 209, the paper mentions that "post-training appears to induce an alternative task inference mechanism." Could the authors clarify what this mechanism is?

Please also refer to the weaknesses

**Ethical Concerns:**

["NO or VERY MINOR ethics concerns only"]

**Limitations:**

yes

**Quality:**

3

**Strengths And Weaknesses:**

Strengths:

- The paper generalizes the function vector approach from in-context demonstrations to instructions.
- The experimental design is comprehensive. Many of the findings are novel and informative to me.
- The paper is well-written and clearly structured, making it easy to follow the authors' arguments.

Weaknesses:

- While the paper reports several interesting empirical phenomena, some are not sufficiently explained or discussed. For example:
  - Why does the Llama-3.1-8B model underperforms  Llama-3.1-3B model (Figure 2)?
  - Is the improvement from combining both demonstration and instruction FVs due to capturing complementary information, or is it merely amplifying the intervention strength?
  - What causes post-training to align instruction FV heads closer to demonstration FV heads, and why does the average layer of instruction-only heads shift from being deeper than that of demonstration heads to approximately the same depth?
- Although the experiments on synthetic tasks provide useful insight into task representation mechanisms, the paper would benefit from more practical takeaways or actionable guidance, for example on model steering in real-world scenarios.

---

> ### Author Rebuttal · Authors · 2025-07-28
>
> ## Response
>
> Thank you for your engagement with our work. We are glad you appreciated our comprehensive set of experiments and findings, as well as the structure of our paper. We respond to your highlighted weaknesses and questions below:
>
> **Weakness 1:**
>
> >While the paper reports several interesting empirical phenomena, some are not sufficiently explained or discussed. For example:
>
> Thank you for highlighting several findings we could articulate or explain better. We offer some answers below, and we commit to adding similar clarifications to the camera-ready version of our work:
>
> >Why does the Llama-3.1-8B model underperforms Llama-3.1-3B model (Figure 2)?
>
> We remark in lines 156-158 that we believe the heuristic L/3 layer in which the intervention is added is highly suboptimal for that model specifically. Figure 11 (Appendix G.1, L972, p. 29) is similar to Figure 2, but plots using the best intervention depth for each model. In this figure, Llama-3.1-8B closes some of the gap.
>
> Another factor that may contribute to this is that the Llama-3.2 series is distilled using logits from both the 8B and 70B Llama-3.1 models, which may confer benefits beyond the 8B model. We will add a note to this effect and clarify the previous one in the camera-ready version.
>
> >Is the improvement from combining both demonstration and instruction FVs due to capturing complementary information, or is it merely amplifying the intervention strength?
>
> We were also curious about this question, so we ran some experiments in which we added a single FV twice, rather than both of them. We summarize our findings in L182-187 and plot them in Appendix G.4 (L975, p. 31). In some models and evaluation settings, adding an FV twice is equivalent to (or marginally better than) adding both FVs, but in many it isn’t. If you think it would be helpful, we are happy to clarify this finding in the camera-ready version.
>
> >What causes post-training to align instruction FV heads closer to demonstration FV heads, and why does the average layer of instruction-only heads shift from being deeper than that of demonstration heads to approximately the same depth?
>
> Thank you for this question. We do not have a complete answer regarding post-training dynamics. Our best guess is that instruction-tuning uses attention heads that are peripherally useful for ICL and leverages them to represent instructed tasks. We explain our thinking slightly later, in L228-232. We will add a note suggesting the connection between this change in depth and this mechanism to the camera-ready version of the paper.
>
> **Weakness 2:**
>
> >Although the experiments on synthetic tasks provide useful insight into task representation mechanisms, the paper would benefit from more practical takeaways or actionable guidance, for example on model steering in real-world scenarios.
>
> Thank you for this comment. We attempted to be cautious in our discussion to avoid overstating the implications of our work, given that the suite of tasks we evaluate is, as you note, relatively simple. One observation our work provides is that instruction task representation is formed across several layers in the middle of the model. Most steering methods intervene at a single layer. Our work suggests that there may be benefits to applying smaller interventions across different layers rather than a single intervention in a particular layer, and we leave this for future work to explore. We can also offer some speculation on how our findings might inform prompt design feedback. One idea we began exploring is monitoring the representation in the instruction function vector heads as a proxy for “has the language model formed a task representation from this prompt?” If we were able to monitor such a signal (e.g., as a function of the outputs of these attention heads), that could provide a signal to the prompt designer indicating that they should refine or rephrase their instructions. A second potentially useful direction might be in safety: what if we could classify whether a model inferred a safe task by training a classifier on the activity in function vector heads? We will discuss such ideas and how to make progress toward them in the camera-ready version of the paper.
>
> **Question 1:**
>
> >Do the authors have an explanation for why FVs facilitate zero-shot accuracy less effectively on the LLaMA-3.1-8B model than on the 3B model? Why does the combination of instruction and demonstration FVs fail to improve performance on the 8B model in the Shuffled 10-shot evaluation setting, even when using the empirically optimal intervention depth (Figure 11)? I'm uncertain whether this indicates that the effectiveness becomes more restricted as model size increases.
>
> Beyond the thoughts offered in response to the first question on the first weakness identified, one other note we can make is that we use the same number of attention heads in both of these models, which are proportionally more in the 3B models (the 3B models have 672 attention heads, so 20 top heads are ~3%, while the 8B models have 1024 attention heads, so the top 20 are ~2%). It could be that we are missing more useful attention heads in the 8B models. We will add a note to this effect in the camera-ready version.
>
> **Question 2:**
>
> >In Line 209, the paper mentions that "post-training appears to induce an alternative task inference mechanism." Could the authors clarify what this mechanism is?
>
> What we meant by ‘alternative’ L209 is that the mechanism induced by post-training for instruction task representations is separate from the one base models develop for demonstration task representation. We have so far identified this mechanism only through a different set of attention heads and do not have a full mechanistic description. We will clarify this in the camera-ready version.
>
>
> ## Camera-ready commitments
> - **Re: Weakness 1:** We commit to adding a few notes:
>   - Regarding the underperforming Llama-3.1-8B, we will clarify our hypothesis about the intervention layer being suboptimal and add a note about the logit distillation used in the creation of the Llama-3.2 models.
>   - Regarding the intervention amplification question, if you think it would be helpful, we are happy to clarify our findings in the camera-ready version of the manuscript.
>   - Regarding the FV head depth alignment question, we will add a note explicitly connecting our hypothesis about what instruction-tuning is doing to the observed changes in alignment depth.
> - **Re: Weakness 2:** Expand the practical future-looking discussion of the downstream utility of our findings. We will add a treatment to our discussion section on what our findings might imply for steering and prompt design, as well as the future work required to validate and implement these ideas, building on the ideas we sketch out in our response above.
> - **Re: Question 1:** We will add a note regarding the fraction of attention heads being different across models to the camera-ready version of the manuscript.
> - **Re: Question 2:** We will add a clarification of what we meant by alternative mechanism to the camera-ready version of the paper.
>
> Thank you again for your engagement with our work. We’re happy to answer any other questions you may have or provide additional clarifications. If you feel our response addresses your concerns, we hope you will consider raising your score.

---

> > ### Comment · Reviewer_KJBU · 2025-08-04
> >
> > I thank the authors for the detailed response. I believe these clarifications will make the camera-ready version clearer. I will maintain my score.

---

> > > ### Comment · Area_Chair_ts3M · 2025-08-04
> > >
> > > Hi, please make sure you confirm that you've read and taken into account to the author response with the Mandatory Acknowledgment button.

---

### Official Review · Reviewer_CJWJ · 2025-07-02

**Clarity:** 3
**Significance:** 2
**Originality:** 2
**Rating:** 4
**Confidence:** 3

**Summary:**

This paper investigates whether different prompting methods elicit the same underlying task representations in LLMs. The authors mostly focused on two primary approaches for in-context learning, namely demonstrations and instructions. The authors conducted the study with function vectors, and found out that demonstrations and instructions do not yield a single shared task representation in LLMs, but rather activate different internal mechanisms.

**Questions:**

Please refer to the weaknesses above.

**Ethical Concerns:**

["NO or VERY MINOR ethics concerns only"]

**Final Justification:**

The authors' rebuttal has addressed my concerns. I am maintaining my rating of borderline accept.

**Limitations:**

yes

**Quality:**

3

**Strengths And Weaknesses:**

### Strengths

* This paper has a clear motivation in conducting this study since the interpretability issue is one of the key problems in LLMs explanability and providing thorough examination over this problem can greatly improve LLMs’ performace on downstream applications.

* The paper is comprehensive in scope, since it conducted extensive experiments across various LLM backbones (Llama-2, Llama-3.1, Llama-3.2, OLMo-2) and multiple model sizes (1B/3B/7B/8B) and tasks (~50 datasets), which increases the generalizability of the findings.


### Weaknesses

* While the paper largely follows the setup of Todd et al. ("Function vectors in large language models"), the datasets used in this study, whether  the Abstractive Tasks or the Extractive Tasks, mostly tend to involve relatively simple or short outputs. It remains unclear whether the findings in this work could be generalize to more complex or open-ended tasks (e.g., summarization, reasoning, longer-text generation, etc).

* This is not a major concern, but the paper would benefit from a more explicit discussion of the broader implications of its findings. For instance, how might the insights about task representation influence practical prompt design for in-context learning in real-world applications? Providing such guidance could enhance the paper’s utility and impact.

---

> ### Author Rebuttal · Authors · 2025-07-28
>
> ## Response
>
> Thank you for your engagement with our work. We’re glad you appreciate the clear motivation and breadth of experiments we conducted. We would like to offer a few thoughts regarding the weaknesses you highlight:
>
> **Weakness 1:**
>
> >While the paper largely follows the setup of Todd et al. ("Function vectors in large language models"), the datasets used in this study, whether the Abstractive Tasks or the Extractive Tasks, mostly tend to involve relatively simple or short outputs. It remains unclear whether the findings in this work could be generalize to more complex or open-ended tasks (e.g., summarization, reasoning, longer-text generation, etc).
>
> Thank you for pointing this out; we agree that we did not assess generalization to longer or more complex tasks. Our emphasis in this work was to extend the function vector method to different prompting methods; therefore, we focused on matching their datasets, rather than evaluating this method on more challenging tasks. We also found simple tasks to be useful for elucidating model mechanisms, as is common in interpretability work (for example, see also Hendel et al.’s work). We also consider the free-response nature of the simple tasks as important, as multiple-choice questions (as in MMLU) don’t leave the nature of the task open to interpretation (a multiple-choice question implies the task in the question, even without any instructions). Still, this is an interesting direction, and we leave it to future work to identify the appropriate domain(s) and further study the generalization of these methods. We will expand the current treatment of this limitation in L333-334 to explicitly note the short-output nature of the tasks we evaluate and the generalizability question.
>
> **Weakness 2:**
>
> >This is not a major concern, but the paper would benefit from a more explicit discussion of the broader implications of its findings. For instance, how might the insights about task representation influence practical prompt design for in-context learning in real-world applications? Providing such guidance could enhance the paper’s utility and impact.
>
> Thank you for this comment. We attempted to be cautious in our discussion to avoid overstating the implications of our work, given that the suite of tasks we evaluate is, as you note, relatively simple. That being said, we are happy to offer a speculative proposal on how our work might inform prompt design and what sort of future work would be required to make progress toward this. One idea we began exploring is monitoring the representation in the instruction function vector heads as a proxy for “has the language model formed a task representation from this prompt?” If we were able to monitor such a signal (e.g., as a function of the outputs of these attention heads), that could provide a signal to the prompt designer indicating that they should refine or rephrase their instructions. A second potentially useful direction might be in safety: what if we could classify whether a model inferred a safe task by training a classifier on the activity in function vector heads? We will discuss such ideas and how to make progress toward them in the camera-ready version of the paper.
>
> ## Camera-ready commitments
> - **Re: Weakness 1:**  Expand the discussion of the task set limitations to emphasize that the tasks used were all minimal and specifically mention the issue of generalization to more complex and open-ended tasks.
> - **Re: Weakness 2:** Expand the future-looking discussion of the downstream utility of our findings. We will add a treatment to our discussion section on what our findings might imply for prompt design and the future work required to validate and implement these ideas, building on the ideas in the response above.
>
> Thank you again for your engagement with our work. We’re happy to answer any other questions you may have or provide additional clarifications. If you feel our response addresses your concerns, we hope you will consider raising your score.
>
> ## References
>
> Hendel, R., Geva, M., and Globerson, A. (2023). In-context learning creates task vectors.

---

### Official Review · Reviewer_25fA · 2025-07-03

**Clarity:** 3
**Significance:** 2
**Originality:** 3
**Rating:** 4
**Confidence:** 3

**Summary:**

In this paper, the authors investigated how different task specification, such as instruction or in-context examples, can infer different task representation. The authors measured the task representations via functional vectors, and identify that different task specification can leads to complementary task representation, which later on can be combined and help steer the models to achieve better performance. Further analysis discuss how different task specification engage different attention heads.

**Questions:**

1. Line 200 attention head analysis: "This suggests that task inference from demonstrations is more localized to a small set of heads than from instructions and that instruction-based task inference is more diffuse". Though this is a finding, what's the further insight of this finding? For example, how does it help future model steering development?
2. In Section 2.1 Generalizing function vectors beyond in-context demonstrations, when constructing uninformative baselines, do you consider generate the negate instructions as the uninformative baseline?
3. In Section 3.5 Fig 4, when we do see using instruction FV of post-trained models for pre-trained models help boost performance, the demonstration FV seems to be not helpful. Does it means that post-training does not help with ICL and the demonstration FV does not become more informative?

**Ethical Concerns:**

["NO or VERY MINOR ethics concerns only"]

**Final Justification:**

The author has addressed my concerns about the evaluation and clarity issue and I'm raising the clarity score from 2 to 3 correspondingly.

**Limitations:**

yes

**Quality:**

3

**Strengths And Weaknesses:**

# Strength
1. This paper discusses the questions of how different task specification can leads to different representation. The fact that they do brings insights towards how LMs process diverse prompts in different ways. This finding aligns with previous observations of instruction fine-tuning with diverse instructions leads to better generalization.
2. The authors propose a simple method to generalize the functional vector constructions from in-context examples to other type of task instruction.
3. The combination of FVs of different task specifications leads to better zero-shot performance, showing the potential of improving model performance with diverse prompts without training. This has the potential to be an useful method for steering models behavior with better performance.
4. The analysis of using FVs from post-training models for pre-training model shows that FV can effectively steer the models latent representation and help boost the performance.

# Weakness
1. The writing can be improved. The authors introduce the background of Functional Vector and the proposed methods with a sequences of mathematical terms without clear structure, which makes it difficult to follow. While I can understand the methods after reading in detail, I'll suggest the authors update the method section with visualization of FV calculations for better clarity.
2. While this paper shows that combining both FVs leads to better performance on a specific dataset used by Todd et al. (2024)., how this methods generalize to other datasets is unclear to me. Does this method also works on other prominent benchmarks, such as MMLU or BBH?

---

> ### Author Rebuttal · Authors · 2025-07-28
>
> ## Response
>
> Thank you for your engagement with our work. We’re glad you appreciate the insights our work provides into how LMs process different types of prompts and the potential for using our findings to develop new steering methods. We respond to your highlighted weaknesses and questions below:
>
> **Weakness 1:**
>
> >The writing can be improved. The authors introduce the background of Functional Vector and the proposed methods with a sequences of mathematical terms without clear structure, which makes it difficult to follow. While I can understand the methods after reading in detail, I'll suggest the authors update the method section with visualization of FV calculations for better clarity.
>
> Thank you for this suggestion. We’ll be happy to add a figure to clarify this computation. We commit to adding a visualization of the function vector calculation to the camera-ready version of the work (under the new rebuttal guidelines, we cannot upload a figure or an updated version, but we commit to adding one to the final camera-ready copy).
>
> **Weakness 2:**
>
> >While this paper shows that combining both FVs leads to better performance on a specific dataset used by Todd et al. (2024)., how this methods generalize to other datasets is unclear to me. Does this method also works on other prominent benchmarks, such as MMLU or BBH?
>
> Thank you for this question. We offer a few thoughts:
> - Please note that we do not test the method on a single dataset, but, following Todd et al., we use approximately 40 separate tasks in our evaluation (see Table 5 in Appendix D).
> - Our emphasis in this work was to extend the function vector method to different prompting methods; therefore, we focused on matching their datasets, rather than evaluating this method on more challenging tasks.
> - We found simple tasks to be useful for elucidating model mechanisms, as is common in interpretability work (for example, see also Hendel et al.’s work). We also consider the free-response nature of the simple tasks as important, as multiple-choice questions (as in MMLU) don’t leave the nature of the task open to interpretation (a multiple-choice question implies the task in the question, even without any instructions). Still, this is an interesting direction, and we leave it to future work to identify the appropriate domain(s) and further study the generalization of these methods. We will expand the current treatment of this limitation in L333-334 and specifically mention MMLU and BBH.
>
> **Question 1:**
>
> >Line 200 attention head analysis: "This suggests that task inference from demonstrations is more localized to a small set of heads than from instructions and that instruction-based task inference is more diffuse". Though this is a finding, what's the further insight of this finding? For example, how does it help future model steering development?
>
> Thank you for this note; we agree we could make this finding more actionable. We will add a note to this effect in the final, camera-ready version of the manuscript: “This implies that interventions focused on changing or improving instruction-following may benefit from intervening at multiple locations and depths in a model than interventions aimed at changing demonstration-following.”
>
> **Question 2:**
>
> >In Section 2.1 Generalizing function vectors beyond in-context demonstrations, when constructing uninformative baselines, do you consider generate the negate instructions as the uninformative baseline?
>
> We did not consider negating the instructions. Thank you for this suggestion. However, there is evidence that language models often fail to understand negation, which might make such a baseline less informative. See Hosseini et al. (2021), Jang et al. (2023), Alhamoud et al. (2025), and Vrabcová et al. (2025) for examples of the phenomenon.
>
> **Question 3:**
>
> >In Section 3.5 Fig 4, when we do see using instruction FV of post-trained models for pre-trained models help boost performance, the demonstration FV seems to be not helpful. Does it means that post-training does not help with ICL and the demonstration FV does not become more informative?
>
> We agree with your assessment. As far as we can tell, there is no evidence that post-training improves performance on demonstration-based ICL (in private correspondence with Todd et al., they noted very similar results between base and post-trained models in their work, as well). If you think it would help, we are happy to make this point explicitly in the camera-ready version of the manuscript.
>
> ## Camera-ready commitments
>
> - **Re: Weakness 1:** Add a figure to visualize function vector calculation.
> - **Re: Weakness 2:** Expand the discussion of the task set limitations to emphasize that the tasks used were all minimal and specifically mention the issue of generalization to other standard benchmarks, such as MMLU and BBH.
> - **Re: Question 1**: Add a more actionable note regarding our finding on the diffuseness of instruction-based inference with a language similar to the example above.
> - **Re: Question 3:** If you think it would help, add a more explicit note that post-training does not improve demonstration-based ICL.
>
> Thank you again for your engagement with our work. We’re happy to answer any other questions you may have or provide additional clarifications. If you feel our response addresses your concerns, we hope you will consider raising your score.
>
> ## References
>
> Alhamoud, K., Alshammari, S., Tian Y., Li, G., Torr, P. Kim, Y., and Ghassemi, M. (2025). Vision-Language Models Do Not Understand Negation.
>
> Hendel, R., Geva, M., and Globerson, A. (2023). In-context learning creates task vectors.
>
> Hosseini, A., Reddy, S., Bahdanau, D., Hjelm, R. D., Sordoni, A., and Courville, A. (2021). Understanding by Understanding Not: Modeling Negation in Language Models
>
> Jang, J., Ye, S., and Seo, M., (2023). Can Large Language Models Truly Understand Prompts? A Case Study with Negated Prompts.
>
> Vrabcová, T., Kadlčík, M., Sojka, P. Štefánik, M., and Spiegel, M. (2025). Negation: A Pink Elephant in the Large Language Models' Room?

---

> > ### Comment · Reviewer_25fA · 2025-08-05
> >
> > I thank the authors for the thorough response. With the author clarifications of W2 and the commitment to add visualization for function vector, I've rais the clarification score correspondingly.

---

### Note · Authors · 2025-08-13

We would like to thank the reviewers for the valuable feedback you have provided us.

We are glad the reviewers appreciate our contributions and the strengths of our work. These include include the clear motivation and importance of the problem we study (25FA, CJWJ, fKfR), our generalization of the function vector approach (25fA, KJBU), and the comprehensive scope of our experiments (CJWJ, KJBU, fKfR).

We look forward to making improvements in the camera-ready version of the manuscript. We detailed our commitments in our response to each reviewer, and we summarize recurring themes below:
- We will expand the discussion of the task set limitations to emphasize the minimality of tasks used.
- We will expand the future-looking discussion of the practical downstream utility of our findings. We will add a treatment to our discussion section on what our findings might imply for prompt design and the future work required to validate and implement these ideas.
- We will add a visualization of the function vector calculation and enrich our discussion of related methods.

Of course, we will also address the other camera-ready commitments noted only in our responses to single reviewers.

We appreciate your engagement with our work and the improvements it enables us to make in our final version of this paper.

---

### Decision · Program_Chairs · 2025-09-17

**Decision:**

Accept (poster)

**Comment:**

This paper studies how different variations of task specification (e.g., in-context learning, direct instruction) might result in different representations of a task internal to a model. Because it is found that variations result in different representations, the authors also propose to make use of this to improve model performance in a form of ensembling by mixing activations from multiple function vectors coming from different different task specifications. Experiments are performed on a task previously defined by Todd et al. (2024).

One missing work is Sclar et al. 2024 (ICLR) which also shows that task specifications (prompt formatting) maintain unique internal representations from one another.